# Drivers of future seasonal cycle changes of oceanic pCO$_2$

M. Angeles Gallego[1], Axel Timmermann[2,3,4], Tobias Friedrich[2], and Richard E. Zeebe[1]

[1]Department of Oceanography, School of Ocean and Earth Sciences and Technology, University of Hawaii at Manoa, Honolulu, Hawaii, USA
[2]International Pacific Research Center, School of Ocean and Earth Sciences and Technology, University of Hawaii at Manoa, Honolulu, Hawaii, USA
[3]Center for Climate Physics, Institute for Basic Science (IBS), Busan, South Korea
[4]Pusan National University, Busan, South Korea

**Correspondence:** Angeles Gallego(mdla@hawaii.edu)

**Abstract.** Recent observation-based results show that the seasonal amplitude of surface ocean partial pressure of CO$_2$ (pCO$_2$) has been increasing on average at a rate of 2-3 $\mu$atm per decade (Landschützer et al., 2018). Future increases of pCO$_2$ seasonality are expected, as marine CO$_2$ will increase in response to increasing anthropogenic carbon emissions (McNeil and Sasse, 2016). Here we use 7 different global coupled atmosphere/ocean/carbon cycle/ecosystem model simulations, conducted as part of the Coupled Model Intercomparison Project Phase 5 (CMIP5), to study future projections of the pCO$_2$ annual cycle amplitude and to elucidate the causes of its amplification. We find, that for the RCP8.5 emission scenario the seasonal amplitude (climatological maximum-minus-minimum) of upper ocean pCO$_2$ will increase by a factor of 1.5 to 3 times over the next 60-80 years. To understand the drivers and mechanisms that control the pCO$_2$ seasonal amplification we develop a complete analytical Taylor expansion of pCO$_2$ seasonality in terms of its four drivers: dissolved inorganic carbon (DIC), total alkalinity (TA), temperature (T) and salinity (S). Using this linear approximation we show that the DIC and T terms are the dominant contributors to the total change in pCO$_2$ seasonality. To first order, their future intensification can be traced back to a doubling of the annual mean pCO$_2$, which enhances DIC and alters the ocean carbonate chemistry. Regional differences in the projected seasonal cycle amplitude are generated by spatially varying sensitivity terms. The subtropical and equatorial regions (40$^o$S-40$^o$N), will experience a $\approx$30-80$\mu$atm increase in seasonal cycle amplitude almost exclusively due a larger background CO$_2$ concentration that amplifies the T seasonal effect on solubility. This mechanism is further reinforced by an overall increase in the seasonal cycle of T, as a result of stronger ocean stratification and a projected shoaling of mean mixed layer depths. The Southern Ocean will experience a seasonal cycle amplification of $\approx$90-120 $\mu$atm in response to the mean pCO$_2$-driven change of the mean DIC contribution and to a lesser extent to the T contribution. However, a decrease of the DIC seasonal cycle amplitude somewhat counteracts this regional amplification mechanism.

## 1 Introduction

Owing to its large chemical capacity to resist changes in CO$_2$ concentration ([CO$_2$]) (referred to as buffering capacity), the ocean has absorbed nearly half of the anthropogenic CO$_2$ produced by fossil fuel burning and cement production since the industrial revolution (Sabine et al., 2004). While the ocean's absorption of CO$_2$ lowers the atmospheric concentration, it also

increases the ocean's $[CO_2]$ and in turn lowers its buffering capacity. This leads to a reduction in the oceanic uptake of $CO_2$ and an intensification of the $pCO_2$ seasonal cycle (from now on referred to as $\delta pCO_2$) (McNeil and Sasse, 2016; Völker et al., 2002). In a recent key observational study by Landschützer et al. (2018), it was demonstrated that the $\delta pCO_2$ amplitude has increased at a rate of $\approx$2-3 $\mu$atm per decade, from 1982 to 2015.

The $pCO_2$ already experiences large seasonal fluctuations, which in some regions can reach up to 60% above and below the annual mean $pCO_2$, (Takahashi et al., 2002). An intensification of the $\delta pCO_2$ amplitude could produce seasonal hypercapnia conditions (McNeil and Sasse, 2016) which, together with increased $[H^+]$ seasonality (Kwiatkowski and Orr, 2018; Hagens and Middelburg, 2016) and aragonite undersaturation events (Hauri et al., 2015; Sasse et al., 2015; Shaw et al., 2013) could expose marine life to harmful seawater conditions earlier than expected if considering only annual mean values. Moreover, a

projected amplification of $\delta pCO_2$ might increase the net $CO_2$ uptake in some regions, such as the Southern Ocean, thereby further accelerating the decrease of the buffering capacity in that region (Hauck and Völker, 2015).

The $pCO_2$ seasonal amplitude is controlled mainly by the seasonal changes in temperature (T) and biological activity together with upwelling changes that alter DIC concentrations. Usually, DIC and T changes work in opposite directions (?Takahashi et al., 2002; Fay and McKinley, 2017). In subtropical regions higher $pCO_2$ values occur in summer when solubility decreases.

In subpolar regions, $pCO_2$ increases in winter when waters upwell that are rich in DIC and when respiration of organic matter takes place. Decreased subpolar $pCO_2$ occurs in summer when the primary productivity is higher and the upwelling diminishes. Therefore, we find close relationships of $\delta pCO_2$ with the ocean's $[CO_2]$ that controls the chemical reactions and with the mean $pCO_2$ that moderates the exchange with the atmosphere. Both factors are related by the solubility constant that depends on temperature and salinity.

Furthermore, the regional differences in the influence of temperature and biology on $\delta pCO_2$ are modulated by the ocean's buffering capacity. This is due to the ability of $CO_2$ to react with seawater to form bicarbonate $[HCO_3^-]$ and carbonate $[CO_3^{2-}]$, leaving only a small portion of the dissolved carbon dioxide in the form of aqueous $CO_2$ ($[CO_2(aq)]$). $[CO_2(aq)]$ together with the carbonic acid ($[H_2CO_3]$) are defined as $[CO_2]$. Therefore, it is useful to define the total amount of carbon as DIC, which is the sum of the three carbon species ($[HCO_3^-]$, $[CO_3^{2-}]$ and $[CO_2]$). At current chemical conditions, most of the

DIC is in form of $HCO_3^-$, therefore the buffering capacity is largely controlled by the $CO_3^{2-}$ capable of transforming $CO_2$ into bicarbonate through the reaction $CO_2(aq) + CO_3^{2-} + H_2O = 2HCO_3^-$ (Zeebe and Wolf-Gladrow, 2001). The larger the buffering capacity, the larger the $pCO_2$'s ability to resist changes in DIC. To quantify this capacity, we can introduce the sensitivity factor $\gamma_{DIC}$, which is inversely related to the buffering capacity, defined as $\gamma_{DIC} = \partial \ln(pCO_2)/\partial DIC$, (Egleston et al., 2010). Other sensitivity factors are related to the total alkalinity ($\gamma_{TA}$), salinity ($\gamma_S$) and temperature ($\gamma_T$) changes, and

are defined in a similar way as $\partial \ln(pCO_2)/\partial TA$, $\partial \ln(pCO_2)/\partial S$ and $\partial \ln(pCO_2)/\partial T$ respectively. It is important to note that the $pCO_2$ is highly sensitive to temperature due to two factors: first through solubility changes that account for 2/3 of the present day temperature impact, and second, through the dissociation constants that control the carbon system reactions (Sarmiento and Gruber, 2006).

While the mechanisms controlling the seasonal cycle of $pCO_2$ at present day are well documented, the future evolution of

these drivers has not been fully elucidated. Current literature suggests that the seasonal amplification is a consequence of an

increase on the T and DIC contributions to $\delta pCO_2$ (Landschützer et al., 2018) and an increased sensitivity of the ocean to these variables (Fassbender et al., 2017).

The aim of our paper is to provide an in-depth analysis of the mechanisms controlling the future strength of $\delta pCO_2$ and its regional differences using 7 CMIP5 global earth system models. Our analysis focuses on the $21^{st}$ century evolution using the

Representative Concentration Pathway 8.5 (RCP8.5) scenario. We give a comprehensive analysis of the projected evolution of the DIC, TA, T and S contributions to $pCO_2$ seasonality. To achieve this goal, we derive explicit analytical expressions for $pCO_2$ sensitivities in terms of $\gamma_{DIC}$, $\gamma_{TA}$, $\gamma_T$ and $\gamma_S$, thereby extending previous work done by Egleston et al. (2010).

## 2   Methodology

### 2.1   CMIP5 Models

For our analysis, $pCO_2$, DIC, TA, T and S monthly-mean output variables covering the period from 2006-2100 were obtained from future climate change simulations conducted with 7 fully coupled earth system models that participated in the Coupled Model Intercomparison Project, Phase 5 (CMIP5). The following models were selected based on data availability: CanESM2, CESM1-BGC, GFDL-ESM2M, MPI-ESM-LR, MPI-ESM-MR, HadGEM2-ES and HadGEM2-CC (See supplementary material of Hauri et al. (2015)). For the purpose of this paper, we used the Representative Concentration Pathway 8.5 (RCP8.5)

future climate change simulations (IPCC, 2013). The ocean's surface data sets were regrided onto a $1^o$x$1^o$ grid using Climate Data Operators (CDO). The Arctic Ocean and the region poleward of $70^o$S are removed from the analyses, because observational data for model validation are scarce.

### 2.2   Analysis of $\delta pCO_2$

To elucidate the underlying dynamical, thermodynamical, biological and chemical processes controlling $\delta pCO_2$ we calculated

a first order Taylor series expansion of $\delta pCO_2$ in terms of its four drivers, DIC, TA, T and S. While T and S are controlled only by physics, DIC and TA are controlled by physical, chemical and biological processes. Throughout this paper we use salinity-normalized DIC and TA using a mean salinity of 35 psu. This effectively removes the concentration/dilution fresh water effect, following the procedure of Lovenduski et al. (2007). The salinity normalized variables are referred to as $DIC_s$ and $TA_s$, corresponding to $DIC \cdot S_0/S$ and $TA \cdot S_0/S$ respectively. The freshwater effect on DIC and TA is now included in the S

term, renamed as $S_{fw}$. For the Taylor series expansion, each variable (X=DIC, TA, T and S) is decomposed into $X = \overline{X} + \delta X$. The term $\overline{X}$ represents the 21 years-long mean and $\delta X$ denotes the seasonal cycle (calculated as the monthly mean deviation from the 21 years average). The Taylor's expansion is then computed for an initial (2006-2026) and final (2080-2100) periods. We use multi-decade means and eventually multi-model ensemble means to remove effects of interannual variability. The full first-order series expansion is given by:

$$\delta pCO_2 \quad \approx \quad \frac{\partial pCO_2}{\partial DIC}\bigg|_{\substack{TA,DIC \\ \overline{T},\overline{S}}} \delta DIC_s + \frac{\partial pCO_2}{\partial TA}\bigg|_{\substack{TA,DIC \\ \overline{T},\overline{S}}} \delta TA_s + \frac{\partial pCO_2}{\partial T}\bigg|_{\substack{TA,DIC \\ \overline{T},\overline{S}}} \delta T + \frac{\partial pCO_2}{\partial S}\bigg|_{\substack{TA,DIC \\ \overline{T},\overline{S}}} \delta S_{fw} \tag{1}$$

Each term of the right hand side of Eq. (1) represents the contribution from one of the four drivers of $\delta$pCO$_2$.

The analytical expressions for the derivatives (without the salinity normalization) are given by:

$$\frac{\partial pCO_2}{\partial TA}\bigg|_{\overline{T},\overline{S}}^{TA,DIC} = \overline{pCO_2} \cdot \frac{-\overline{Alk_c}}{\overline{DIC} \cdot \Theta - \overline{Alk_c}^2} \qquad (2)$$

$$\frac{\partial pCO_2}{\partial DIC}\bigg|_{\overline{T},\overline{S}}^{TA,DIC} = \overline{pCO_2} \cdot \frac{\Theta}{\overline{DIC} \cdot \Theta - \overline{Alk_c}^2}$$

$$\frac{\partial pCO_2}{\partial T}\bigg|_{\overline{T},\overline{S}}^{TA,DIC} = \overline{pCO_2} \cdot \frac{1}{\overline{DIC} \cdot \Theta - \overline{Alk_c}^2}\left[\overline{TA_c} \cdot \left(\frac{\partial Alk_c}{\partial T} + \frac{\partial [B(OH)_4^-]}{\partial T} + \frac{\partial [OH^-]}{\partial T}\right) - \Theta \cdot \frac{\partial (DIC - [CO_2])}{\partial T}\right] - \frac{\overline{pCO_2} \cdot}{\overline{K_0(T,S)}}\frac{\partial K_0(T,S)}{\partial T}$$

$$\frac{\partial pCO_2}{\partial S}\bigg|_{\overline{T},\overline{S}}^{TA,DIC} = \overline{pCO_2} \cdot \frac{1}{\overline{DIC} \cdot \Theta - \overline{Alk_c}^2}\left[\overline{Alk_c} \cdot \left(\frac{\partial \overline{Alk_c}}{\partial S} + \frac{\partial [B(OH)_4^-]}{\partial S} + \frac{\partial [OH^-]}{\partial S}\right) - \Theta \cdot \frac{\partial (DIC - [CO_2])}{\partial S}\right] - \frac{\overline{pCO_2} \cdot}{\overline{K_0(T,S)}}\frac{\partial K_0(T,S)}{\partial S}$$

where $\Theta = [\text{HCO}_3^-] + 4[\text{CO}_3^{2-}] + \frac{[\text{B(OH)}_4^-][\text{H}^+]}{(\text{k}_b + [\text{H}^+])} + [\text{H}^+] + [\text{OH}^-]$ and $\overline{\text{Alk}_c} = [\text{HCO}_3^-] + 2[\text{CO}_3^{2-}]$. The explicit T and S partial derivatives are given in the Supplementary material (Text S1). The first two derivatives coincide with the results of Egleston et al. (2010) and Hagens and Middelburg (2016), with the exception of the sign of $[OH^-]$ in Egleston et al. (2010) term $S$. To verify this approach we compared the sum of the Taylor expansion terms with the full simulated range of $\delta$pCO$_2$ from the model's output. The Taylor expansion reproduces well the full seasonal cycle amplitude of the original climate model simulations (Supplementary Fig. S1). The analytical expressions for temperature and salinity presented in here are – to our knowledge – the first ones of their kind. Previously the calculation of these terms was based on the approximation given by Takahashi et al. (1993) or on numerical calculations.

To gain more insight into the processes causing the amplification of $\delta$pCO$_2$ we introduce a method based on a second Taylor series expansion described below. Eq. (1) can be rewritten using the expressions for the sensitivities $\gamma$ determined by the relation $\frac{1}{\text{pCO}_2}\frac{\partial \text{pCO}_2}{\partial X} = \gamma_X$. These sensitivities have been historically used to represent the percentage of change in pCO$_2$ per unit of DIC, TA, T or S. With this notation, Eq. (1) can be expressed in the following way:

$$\delta pCO_2 \approx \overline{pCO_2} \cdot \left(\gamma_{DIC} \cdot \delta DIC_s + \gamma_{TA} \cdot \delta TA_s + \gamma_T \cdot \delta T + \gamma_{S_{fw}} \cdot \delta S_{fw}\right) \qquad (3)$$

Each term in Eq.(3) consists of three parts: $\overline{\text{pCO}_2}$, the sensitivity $\gamma_X$ and the corresponding seasonal cycle $\delta$X. To understand which component is the main driver for $\delta$pCO$_2$ changes, we perform a second Taylor expansion of the end of the century's $\delta$pCO$_2$ around the initial state of the system in 2006-2026.

To maximize mathematical clarity we will introduce some definitions: first, we introduce the symbol $\Delta$ to indicate the difference between the period 2080-2100 and 2006-2026. Therefore, the total future change in $\delta$pCO$_2$, is now referred to as $\Delta\delta$pCO$_2$. In the same manner, the total change in sensitivities and seasonal cycles are written as $\Delta\gamma_{\text{DIC}_s}, \Delta\gamma_{\text{TA}_s}, \Delta\gamma_{\text{T}}, \Delta\gamma_{\text{S}_{\text{fw}}}$, and $\Delta\delta\text{DIC}_s, \Delta\delta\text{TA}_s, \Delta\delta\text{T}, \Delta\delta\text{S}_{\text{fw}}$ respectively. Finally, we introduce the vector $\boldsymbol{X}$ formed by the four variables DIC$_s$, TA$_s$,

T and $S_{fw}$, as: $\{X_0, X_1, X_2, X_3\} = \{DIC_s, TA_s, T, S\}$. With this notation, we can write an expansion of Eq.(3) of the final state of the system by 2080-2100 named $\mathbf{X}^f$ around the initial state $\mathbf{X}^i = \{DIC_s{}^i, TA_s{}^i, T^i, S_{fw}^i\}$ by 2006-2026 as:

$$
\begin{aligned}
\Delta\delta pCO_2 \quad = \quad & \Delta\overline{pCO_2}\sum_{k=0}^{3}\gamma_{X_k}{}^i \cdot \delta X_k{}^i \\[2mm]
+ \quad & \overline{pCO_2^i}\sum_{k=0}^{3}\Delta\gamma_{X_k} \cdot \delta X_k{}^i \\[2mm]
+ \quad & \overline{pCO_2^i}\sum_{k=0}^{3}\gamma_{X_k}{}^i \cdot \Delta\delta X_k \\[2mm]
+ \quad & \Delta\overline{pCO_2}\sum_{k=0}^{3}\Delta\gamma_{X_k} \cdot \delta X_k{}^i \quad (2^{nd} order \quad terms) \\[2mm]
+ \quad & \Delta\overline{pCO_2}\sum_{k=0}^{3}\gamma_{X_k}{}^i \cdot \Delta\delta X_k \\[2mm]
+ \quad & \overline{pCO_2^i}\sum_{k=0}^{3}\Delta\gamma_{X_k} \cdot \Delta\delta X_k \quad\quad\quad,
\end{aligned}
\tag{4}
$$

where the first, second and third terms represent the contributions to $\Delta\delta pCO_2$ due to changes in the mean $pCO_2$ ($\Delta\overline{pCO_2}$), the $pCO_2$ sensitivities ($\Delta\gamma_{X_k}$) and the seasonal cycles ($\Delta\delta X_k$) respectively; the fourth to sixth rows are the second order terms. This method is similar to the one used by Landschützer et al. (2018).

## 3 Results and discussion

### 3.1 $\delta pCO_2$ amplification

Figure 1, (a) shows the ensemble mean $\delta pCO_2$ amplitude (calculated as climatological maximum-minus-minimum) for the initial period 2006-2026. The values range from $\approx$98 $\mu$atm for the high latitudes (40°S-70°S, 40°N-60°N) to $\approx$60 $\mu$atm between 40°S-40°N. The ensemble mean initial seasonal amplitude range is in good agreement with observational estimates calculated for the reference year 2005 (Takahashi et al., 2014b), and for the 1982-2015 period (Landschützer et al., 2017). The agreement between models and observations is remarkably good in the equatorial regions, but the initial amplitude is slightly overestimated in the mid and high latitudes (see Supplementary Fig. S3).The higher amplitude in models than observations is expected, as the initial period 2006-2026 already experienced an amplification compared to previous years. Moreover, Tjiputra et al. (2014) found that the ocean's $pCO_2$ historical trend is larger in models than observations when it is estimated in large scale areas of the ocean. However, they found that models' $pCO_2$ trends agree with observations when the trends are subsampled to the locations where the observations were taken, and therefore they do a good job reproducing well-known time series. Moreover, differences are expected as Pilcher et al. (2015) suggested that CMIP5 models perform well in reproducing the seasonal cycle timing, but still show considerable errors in reproducing the seasonal amplitude of $pCO_2$ due to differences

in the mechanisms represented in each model, especially in subpolar biomes.

By 2080-2100 the annual cycle amplitude attains values of $\approx197$ $\mu$atm and $\approx101$ $\mu$atm in the high and mid-low latitudes respectively (Fig. 1,(b)). These seasonal variations correspond to 20% and 18% of annual $\overline{\mathrm{pCO_2}}$ for the initial and final periods respectively. Figure 1, (c), shows that the global ocean $\delta\mathrm{pCO_2}$ will intensify by a factor of 1.5 to 3 times for the 2080-2100 period relative to the 2006-2026 reference period. Figure 1, (d), shows the difference in amplitude ($\Delta\delta\mathrm{pCO_2}$); this pattern differs from the ratio, because the ratio overestimates the amplification in areas where the initial amplitude is lower than $\approx10$ $\mu$atm. McNeil and Sasse (2016) used observations and a neural-network-clustering algorithm to project that by year 2100, the $\delta\mathrm{pCO_2}$ amplitude in some regions could be up to ten times larger than it was in year 2000. Our mean amplification factor estimation agrees with the mean threefold amplification found for most of the ocean by McNeil and Sasse (2016). However the high values in this previous study can not be reproduced here - mainly because we consider 21 years average ratios instead of single year ratios, which are strongly affected by interannual variability. Using observations, Landschützer et al. (2018) found an increase of 2.2 $\mu$atm per decade, which is smaller than our findings of a total 42 $\mu$atm increase by the end of the century between 40$^{\mathrm{o}}$S-40$^{\mathrm{o}}$N, and a global-mean change of 81 $\mu$atm on the high latitudes. This difference is again possibly due the higher mean pCO$_2$ values in models than observations.

The global ocean mean amplification factor of $\delta\mathrm{pCO_2}$ roughly coincides with a doubling of $\overline{\mathrm{pCO_2}}$ (Fig. 2). The direct relationship between these two is explained in section 3.5. Figure 1 (e-h) shows the zonal mean panels of (a-d); In general, towards the end of the century the pCO$_2$ amplifies more in high latitudes, but so does the standard deviation uncertainty among models. This regional pattern agrees with the observation-based findings of Landschützer et al. (2018) which show that high latitudes have already experienced a larger amplification than mid-low latitudes from 1982 to 2015. Furthermore, the same pattern is projected by CMIP5 models for the seasonal amplification of [H$^+$] by the end of the century (Kwiatkowski and Orr, 2018). This is expected from the near-linear relation between pCO$_2$ and [H$^+$]. These regional differences in amplification for pCO$_2$ can be explained in terms of the relative magnitudes and the phases between the DIC, TA, T and S contributions, which are explained in subsequent sections.

### 3.2  Present and future drivers of $\delta\mathrm{pCO_2}$

To understand the driving factors of $\delta\mathrm{pCO_2}$ and its spatiotemporal differences, we split $\delta\mathrm{pCO_2}$ into the four different contributions from DIC$_s$, TA$_s$, T and S$_{\mathrm{fw}}$ for the initial and final periods, following Eq. (1). The results are shown in Fig. 3. For most of the ocean, the ensemble mean estimated contributions from DIC$_s$ and T to the present-day $\delta\mathrm{pCO_2}$ are in good agreement with the data-based estimates of Takahashi et al. (2014b) and Landschützer et al. (2017), particularly in the equatorial regions (see Supplementary Fig. S3). However our T and DIC contributions are slightly larger in mid and high latitudes, for the same reasons the pCO$_2$ seasonal amplitude is overestimated (see Section 3.1). Also, differences arise between our DIC$_s$ contribution and the observation-based so called "non-thermal" contribution, because the non-thermal contribution also includes the total alkalinity and salinity effects. Nonetheless, between 40$^{\mathrm{o}}$S-40$^{\mathrm{o}}$N our ensemble mean shows that $\delta\mathrm{pCO_2}$ is dominated by changes in temperature that control CO$_2$ solubility, which decreases in summer enhancing pCO$_2$, in agreement with observations. The Southern Ocean is controlled by DIC, that responds to changes in upwelling and phytoplankton blooms. Both mechanisms act

together to decrease (increase) DIC in summer (winter) (Sarmiento and Gruber, 2006).

The models show that the $\delta pCO_2$ in the 40°N to 60°N band is controlled by T, which disagrees with the above mentioned observations that show a non-temperature dominance in this band. The difference between models and observations arises from two regions: the North Atlantic basin and the North Western Pacific; specifically near the Oyashio Current, and the out-

flows from the Okhotsk Seas (see Supplementary Fig. S3). Most models show a T dominance in the North Atlantic basin; only CESM1-BGC and GFDL-ESM2M show a DIC dominance (see Supplementary Fig. S4). The North Atlantic is one of the major sinks of anthropogenic $CO_2$, however some models fail to estimate its uptake capacity (Goris et al., 2018). Goris et al. (2018) found that models with an efficient carbon sequestration present a DIC-dominated $pCO_2$ seasonal cycle in the North Atlantic, but models with low anthropogenic uptake show a T dominance in this region. In the North-Western Pacific,

Mckinley et al. (2006) found that coarse models are not able to capture the intricate oceanographic features of this area, and therefore the $pCO_2$ seasonality is not well captured.

Towards the end of the century (Fig. 3, right column), the amplification of $\delta pCO_2$ is caused by an increase in the $DIC_s$ and T contributions, and to a lesser extent due to $TA_s$ and $S_{fw}$. Only in the high latitudes the $TA_s$ contribution reinforces the $DIC_s$ effect. The $\delta DIC_s$ and $\delta T$ relative phase and magnitude play an important role in causing regional differences of future $\delta pCO_2$.

For example, between 40°-60°, we find a lower amplification factor than at 30°-40° in both hemispheres (Fig. 1, (c)), contrary to what we expected from the general observed larger amplification at higher latitudes. In this band of lower amplification, the warm water from subtropical regions meets the nutrient rich water from the subpolar regions, but the $DIC_s$ and T effects are almost 6 months out of phase, and therefore their cancellation is larger than in the 30°-40° latitude band; where for example, in the North Atlantic, there is 9 month phase-difference between both contributions. A clear illustration of this phase effect is

found in the Supplementary information (Fig. S5).

In the Southern Ocean there is a shift in the maximum $\delta pCO_2$ occurring from August-September to March-April (Fig. 3, last row). This shift is generated because the T contribution gains importance over $DIC_s$, due to a reduction of $\delta DIC_s$ magnitude at the same time that $\delta T$ increases (Fig. 5). In the Equatorial Pacific region (Fig. 5), T dominates over $DIC_s$ but both contributions are small due to their low seasonality. Therefore, this region will experience a low amplification in $\delta pCO_2$. In this region some

models underestimate the $pCO_2$ trend (Tjiputra et al., 2014), and therefore the seasonal amplification might be underestimated too. In the following sections we conduct further analysis by decomposing each contribution as the result of three factors: the mean $pCO_2$ ($\overline{pCO_2}$), the regional $pCO_2$ sensitivities ($\gamma_{DIC}$, $\gamma_{TA}$, $\gamma_T$ and $\gamma_{S_{fw}}$) and the seasonal cycles ($\delta DIC_s$, $\delta TA_s$, $\delta T$ and $\delta S_{fw}$) as determined in Eq. (3).

### 3.3    Future pCO$_2$ sensitivities

The $\gamma_{DIC}$ and $\gamma_{TA}$ are projected to increase by the end of the century due to a lower ocean buffering capacity produced by increasing temperature and larger background concentrations of DIC (Fassbender et al., 2017). This agrees with our results shown in Fig. 4, which shows that all regions will experience an increase in $\gamma_{DIC}$ and $\gamma_{TA}$. Lower buffer factors (higher sensitivities factors) are found in regions where DIC and TA have similar values, and they will decrease (increase) as the DIC/TA ratio in the oceans increases (Egleston et al., 2010). The alkalinity sensitivity is negative, as $pCO_2$ decreases with

increasing alkalinity, but we show here the negative of $\gamma_{\text{TA}}$ for better comparison. $\gamma_{\text{TA}}$ will increase (with negative values) more than the DIC sensitivity. However seasonal changes in open-ocean $\text{TA}_{\text{s}}$ are small, and therefore the total contribution of alkalinity in our analysis is negligible compared to the $\text{DIC}_{\text{s}}$ and T contributions. $\gamma_{\text{S}_{\text{fw}}}$ decreases everywhere except in the Western Pacific Warm Pool. In this region $\gamma_{\text{S}_{\text{fw}}}$ increases probably due to future changes in precipitation that enhance the

fresh-water effect. In Fig. 4, the sensitivities ($\gamma$) are expressed as a percentage change of pCO$_2$ per unit in DIC, TA, T and S respectively. This follows the approach of Takahashi et al. (1993), however in their paper the authors compute the Revelle factor, which is related to $\gamma_{\text{DIC}}$ as $\text{R} = \text{DIC} \cdot \gamma_{\text{DIC}}$. To illustrate the meaning of the sensitivities, we will focus on the subtropical North Pacific in the 15$^{\text{o}}$N-40$^{\text{o}}$N latitudinal band. In this region $\gamma_{\text{DIC}}$ indicates an average 0.6% change in pCO$_2$ per unit of DIC in 2006-2026. Therefore, for a $\delta\text{DIC}_{\text{s}}$ seasonal cycle amplitude of 40 $\mu$mol/kg$^{-1}$ and $\overline{\text{pCO}_2} \approx 400$ $\mu$atm, the total $\delta$pCO$_2$

amplitude equals 96 $\mu$atm. Following the same reasoning, by 2080-2100, $\gamma_{\text{DIC}}$ increases to 0.7% and $\delta\text{DIC}_{\text{s}}$ decreases to 30 $\mu$mol/kg$^{-1}$; therefore, for a $\overline{\text{pCO}_2}$ equal to 800 $\mu$atm, the $\delta$pCO$_2$ amplitude due to $\delta$DIC amounts to 168 $\mu$atm.

The temperature sensitivity has been experimentally determined by Takahashi et al. (1993); who found a value of 0.0423, meaning that pCO$_2$ changes by about 4% for every $^{\text{o}}$C. This value agrees with our global mean ensemble estimate of 0.0428. However, our analytical expression of $\gamma_T$ shows that this value varies regionally and, by reasons unknown to us, it might

decrease in the future to a global mean value of 0.0415, (Fig. 4, row (c), third column). The T sensitivity is larger in colder regions and lower in the warmer tropics; however, colder regions will experience a larger reduction on $\gamma_{\text{T}}$, which locally prevents a larger amplification of the T contribution to $\delta$pCO$_2$. In the next section we show that the T seasonality is projected to increase in high latitudes, strengthening the T contribution.

### 3.4   Future $\delta$DIC$_{\text{s}}$, $\delta$TA$_{\text{s}}$, $\delta$T and $\delta$S$_{\text{fw}}$.

Towards the end of the century, the global mean amplitude of $\delta\text{DIC}_{\text{s}}$ is projected to decrease by $\approx$26-28% in the high latitudes (Fig.5, (a)), according to all the CMIP5 earth system model simulations used here. In the mid-low latitudinal band there is no agreement between models; while some show an increase others project a decrease in amplitude. As suggested by Landschützer et al. (2018), the larger decrease in the Southern Ocean may be the result of changes in the shallow overturning circulation that prevent CO$_2$ accumulation in this region. This reduction may be counteracted by the predicted increase in productivity owing

to a suppression of light and temperature limitations (Steinacher et al., 2010; Bopp et al., 2013).

According to the CMIP5 models, most of the ocean is projected to experience a slight increase in $\delta$T, as shown in Fig. 5, column (b). All models show a slight increase in $\delta$T, only one model showed a slightly decrease in the southern region, and two models showed a decrease in the equatorial region during October to December. It is important to note that Fig. 5 shows the seasonal values, with the mean T removed. Therefore, when considering the positive T trends, the absolute summer values show an

increase and the absolute winter values a decrease. This agrees with the results of Alexander et al. (2018); who showed that models project a seasonal intensification of T, with larger warm extremes and reduced cold extremes. The authors attributed the T seasonality intensification to an increased oceanic stratification and an overall shoaling of the mixed layer depth, which confines seasonal changes in a reduced volume of water, producing larger changes at the surface. They also showed that the intensification trends are stronger in summer than winter, as the mixed layer depth is shallower in summer. Moreover, ice

covered regions will experience the largest increase in T seasonality due the loss of sea ice, because the ice melting/freezing moderates the surface water temperature seasonality(Carton et al., 2015).

The TA seasonality is also projected to increase in the high latitudes according to all models, except CESM1-BGC which shows a decrease. For $\delta S$ (see Supplementary Fig. S6) there is no agreement among the different CMIP5 models, except in the

Southern Ocean where all the models show a slightly decrease. Kwiatkowski and Orr (2018) demonstrated that the seasonality of the drivers is important to determine future changes in $[H^+]$ seasonality. In the same fashion, our results show that the four $\delta pCO_2$ drivers present changes in seasonality, and in particular $\delta DIC_s$ and $\delta T$ changes are important to explain future projections of the $\delta pCO_2$ amplitude. The increase in $\delta T$ enhances the $\delta pCO_2$ amplification, and the reduction of $\delta DIC_s$ in the Southern Ocean locally prevents a larger amplification.

## 3.5  Regional dominant factors

To identify the main cause of the $\delta pCO_2$ amplification we use the Taylor series expansion method. With this method we consider the system's final state ($\delta pCO_2$ by 2080-2100) as a perturbation of the initial state ($\delta pCO_2$ by 2006-2026), as shown in Eq. (4). The expansion is done in three groups of variables: the seasonal cycles of $DIC_s$, $TA_s$, T and S ($\delta X$) , the sensitivities of $pCO_2$ to the same four variables ($\gamma_x$), and the mean $pCO_2$ ($\overline{pCO_2}$). Therefore, each term of the expansion represents how

much of the total $\delta pCO_2$ change (indicated by $\Delta\delta pCO_2$ and calculated as 2080-2100 value- minus-2006-2026 value) is due the change in each of these factors. We also add the second order terms that come from their combination. The results are shown in Fig. 6, (a) and they indicate that the leading cause of the $\delta pCO_2$ amplification is the change in $\overline{pCO_2}$ ($\Delta\overline{pCO_2}$), which confirms previous findings by Landschützer et al. (2018).

It is important to note that our linear Taylor's expansion approach neglects one aspect of the highly non linear carbonante

chemistry of the ocean: it assumes $\overline{pCO_2}$ and the sensitivities as independent variables, and therefore does not include the positive feedback between larger $\overline{pCO_2}$ and increasing $\gamma_{DIC}$ (decreasing buffering capacity). Hence in the following, we use changes in $\overline{pCO_2}$ and changes in seawater carbonate chemistry synonymously, overall resulting in an enhanced response of $\delta pCO_2$ to seasonal changes in DIC, TA, T and S.

Considering regional differences, we note that the amplification increases as we move poleward in spite of decreasing $\Delta\overline{pCO_2}$

(see Fig. 1 and 2 ). This characteristic geographical pattern of stronger high latitude amplification is the result of larger present-day sensitivities ($\gamma_{DIC_s}$, $\gamma_T$) and seasonal amplitudes ($\delta DIC_s$, $\delta T$) in the high latitudes that amplify the effect of $\Delta\overline{pCO_2}$ even when its value is small compared to other regions (see Eq. (4), first row term). Some exceptions can be found south of Greenland and near the subtropical gyres, where $\Delta\overline{pCO_2}$ reaches higher values and therefore they also present large amplification. We also found spatial differences on smaller scales; for example, the western Equatorial Pacific presents lower initial $\delta pCO_2$ and

amplification than the eastern Equatorial Pacific (see Fig. 1). This is because the eastern side of the basin has larger $DIC_s$ and T contributions than the western side (see Supplementary Fig. S2), as consequence of the upwelling of cold, $CO_2$-rich waters in the east, which lower the buffering capacity and induce larger $\delta pCO_2$ amplitude due the seasonal effects of productivity and solubility (Valsala et al., 2014).

To further disentangle which of the two main drivers ($DIC_s$ or T) is most affected by $\Delta\overline{pCO_2}$, we decomposed the $DIC_s$ and T

contributions in their sensitivity, seasonal cycle and $\overline{\mathrm{pCO_2}}$ components. Figure 6, (b), shows the total DIC and T components together with the $\Delta\overline{\mathrm{pCO_2}}$ and seasonal cycles effects on them. The effects from the sensitivities are not depicted, as they only play a minor role. Only the $\Delta\gamma_{\mathrm{DIC}}$ term gains importance in the Southern Ocean (not shown). In most of the ocean, the $\Delta\overline{\mathrm{pCO_2}}$ effect on T contribution is the leading cause of amplification. This effect is the result of seasonal solubility changes acting over a larger $[CO_2]$ (Gorgues et al., 2010). In the northern high latitudes, an increase on $\delta$T reinforces the amplification. In general, the $\Delta\delta$T contribution gains importance as we move poleward in both hemispheres and therefore the second order terms originating from $\Delta\overline{\mathrm{pCO_2}} \cdot \Delta\delta$T also reinforce the amplification. Interestingly, in the high latitudes, the amplification through second order terms is as important as the change in the seasonality of the drivers.

The Southern Ocean is an exception to the T dominance; in this region the $\Delta\overline{\mathrm{pCO_2}}$ effect on the $\mathrm{DIC_s}$ contribution dominates, and the regional amplification is reinforced by low values of the mean buffering capacity (high $\gamma_{\mathrm{DIC_s}}$). This result agrees with the findings of Hauck and Völker (2015). In this area the amplification is counteracted by a reduction in $\delta\mathrm{DIC_s}$.

## 4 Conclusions

In this study, we used output from 7 CMIP5 global models, subjected to the RCP8.5 radiative forcing scenarios, to provide a comprehensive analysis of the characteristics and drivers of the intensification of the seasonal cycle of pCO$_2$ between present (2006-2026) and future (2080-2100) conditions. By 2080-2100 the $\delta$pCO$_2$ will be 1.5-3 times larger compared to 2006-2026. The projected amplification by the earth-system models and the possible causes of it, are consistent with observation-based amplification for the period from 1982 to 2015 (Landschützer et al., 2018). However, the models slightly overestimate the present day amplification, probably due the larger pCO$_2$ trends in models than observations (Tjiputra et al., 2014).

The models confirm the well-established mechanisms controlling present-day $\delta$pCO$_2$ (Takahashi et al., 2002; Sarmiento and Gruber, 2006; Fay and McKinley, 2017). DIC$_s$ and T contributions are the main counteracting terms dominating the seasonal evolution of $\delta$pCO$_2$. Furthermore, the models show that under future conditions the controlling mechanisms remain unchanged. This result confirms the findings of Landschützer et al. (2018) that identified the same regional controlling mechanism for the past 30 years. The relative role of the DIC and T terms is regionally dependent. High latitudes and upwelling regions, such as the California Current system and the coast of Chile, are dominated by DIC$_s$ and the temperate low latitudes are driven by T. Only in the North Atlantic and North-Western Pacific the models show a dominance of thermal effects over non-thermal effects, which is in disagreement with observations. This further illustrates the urgent need for models to accurately represent regional oceanographic features to accurately reproduce the $\delta$pCO$_2$ characteristics.

In agreement with Landschützer et al. (2018), also the model projections towards the end of this century demonstrate that the global amplification of $\delta$pCO$_2$ is due to the overall longterm increase of anthropogenic CO$_2$. A higher oceanic background CO$_2$ concentration enhances the effect of T-driven solubility changes on $\delta$pCO$_2$ and alters the seawater carbonate chemistry, also enhancing the DIC seasonality effect. The spatial differences of $\delta$pCO$_2$ amplification, however, are determined by the regional sensitivities and seasonality of pCO$_2$ drivers. For example, polar regions show larger sensitivity to DIC and T and larger seasonal cycles of DIC and T. Therefore, these areas present a strong enhancement of $\delta$pCO$_2$, in spite of smaller changes

in mean $pCO_2$.

Moreover, the $pCO_2$ seasonal cycle amplitude depends on the relative magnitude and phase of the contributions. The models ensemble mean reproduces the highly effective compensation of $DIC_s$ and T contributions when they are six months out of phase, confirming previous studies (Takahashi et al., 2002; Landschützer et al., 2018). The compensation of DIC and T prevents

a larger amplification of $\delta pCO_2$, even when both contributions are largely amplified.

The amplification of the TA and S contributions have a small impact on $\delta pCO_2$ in most regions, except in the high latitudes where the TA contribution complements the DIC one, enhancing the non-thermal effect in this region.

The use of earth system models allowed us to state the importance of including future changes on the drivers' seasonalities for future $\delta pCO_2$ projections. The T seasonality is projected to increase in most of the ocean basins, thereby reinforcing the

$\delta pCO_2$ amplification. The $\delta T$ increase is consistent with an increase in stratification that will confine the seasonal changes in net heat fluxes to a shallower mixed layer (Alexander et al., 2018). The $DIC_s$ seasonality decreases in some cold areas and its reduction prevents a larger amplification. For the sensitivities, while $\gamma_{DIC}$ increases, $\gamma_T$ decreases. The later phenomenon needs further study.

The increasing amplitude of $\delta pCO_2$ might have implications for the net air-sea flux of $CO_2$, in particular in regions where

there is an imbalance between winter and summer values (Gorgues et al., 2010). Examples of such behavior can be found in the Southern Ocean (between 50°S-60°S) (Takahashi et al., 2014a) and in the latitude band from 20°-40° in both hemispheres (Landschützer et al., 2014). Moreover, seasonal events of high $pCO_2$ could have an impact on acidification and aragonite undersaturation events (Sasse et al., 2015) and hypercapnia conditions (McNeil and Sasse, 2016). Therefore, understanding the drivers of future $\delta pCO_2$ may help to better assess the response of marine ecosystems to future changes in carbonate chemistry.

Finally, our complete analytical expansion of $\delta pCO_2$ in terms of all its 4 variables provides a practical tool to accurately and quickly diagnose temperature and salinity sensitivities from observational or modelling datasets.

## 5   Acknowledgements

This work was supported by the National Science Foundation under Grant No. 1314209 and by the Institute for Basic Science (IBS), South Korea under IBS-R028-D1. We thank Peter Landschützer for kindly providing his data-sets of thermal and non-

thermal components of the $pCO_2$ seasonal cycle.

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

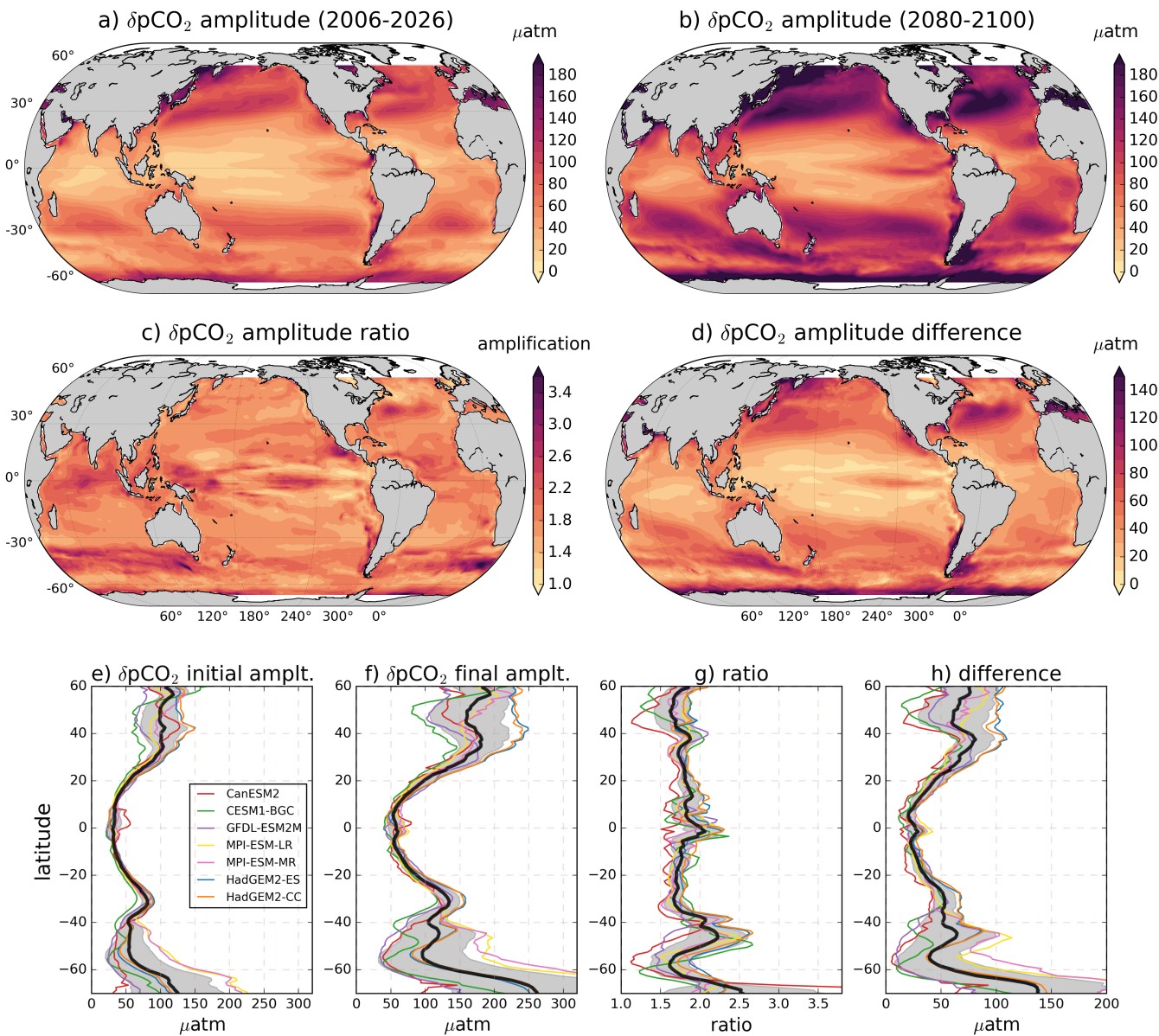

**Figure 1. RCP8.5 ensemble mean pCO$_2$ seasonal cycle amplitude.** Amplitude is calculated as climatology maximum-minus-minimum; for a) initial (2006-2026) and b) final (2080-2100) periods. Initial and final climatologies were calculated as the monthly deviation from the respective 21 years period mean. c) and d) show the ratio and difference between the $\delta$pCO$_2$ amplitudes for 2080-2100 and 2006-2026 respectively. e) - h) show the zonal mean of a)- d) respectively, with the individual models shown as colored lines and the ensemble mean overlaid in black. Gray shading represents one standard deviation across the models.

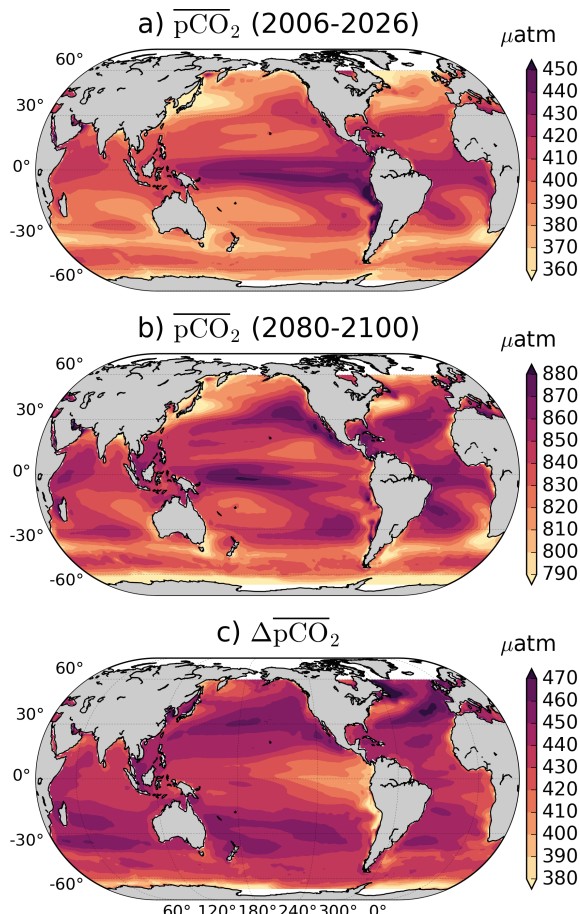

**Figure 2. RCP8.5 ensemble mean $\overline{\text{pCO}_2}$:** by a) 2006-2026 and b) 2080-2100. c) Difference between 2080-2100 and 2006-2026. The North Atlantic and subpolar gyres, show the largest difference between initial and final periods. The scale is different in each plot to enhance regional features.

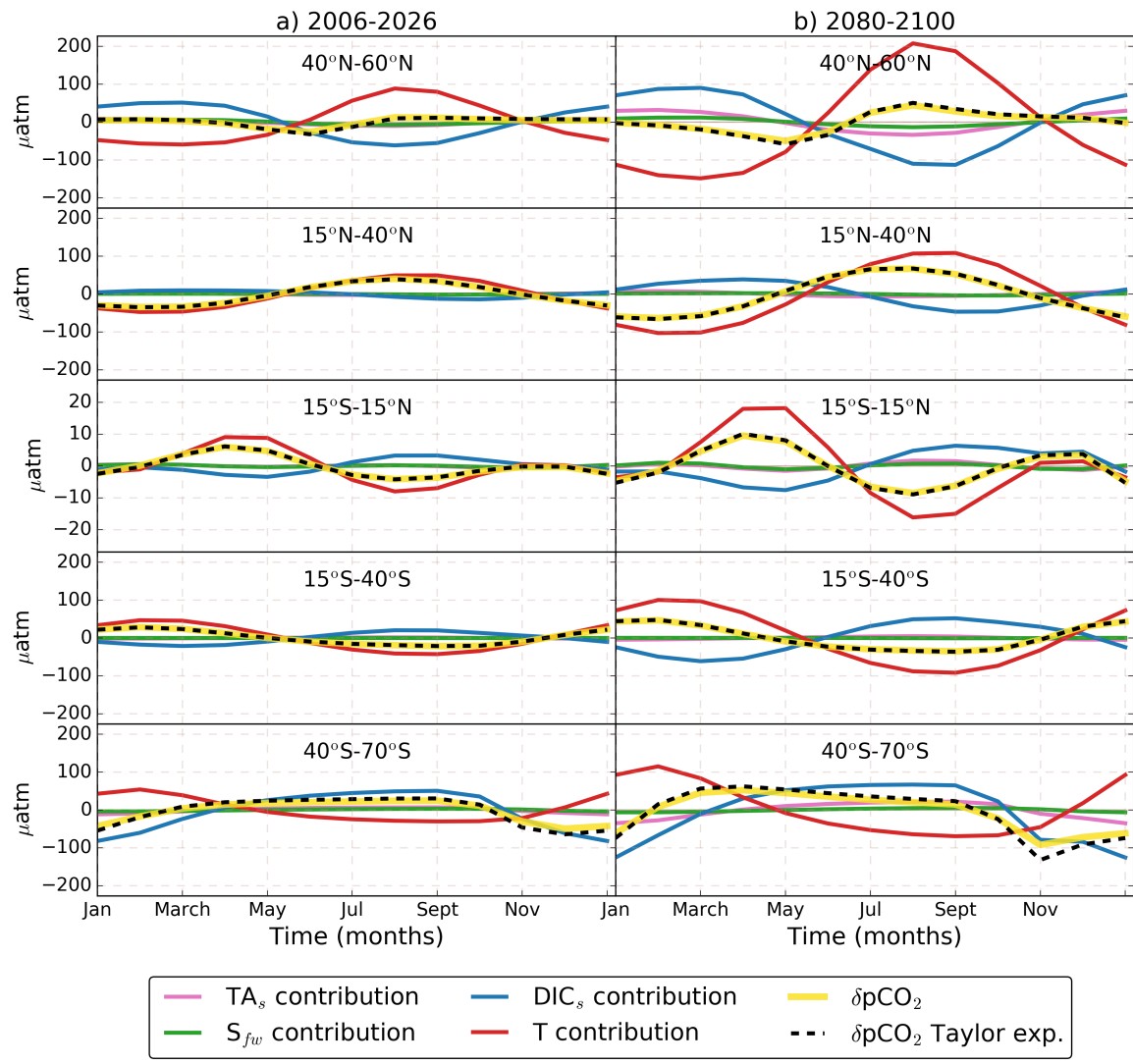

**Figure 3. RCP8.5 ensemble mean seasonal cycle ($\delta$pCO$_2$) and its Taylor decomposition**. Colored lines indicate the contributions of DIC$_s$ (blue), TA$_s$ (pink), T (red) and S$_{fw}$ (green) to $\delta$pCO$_2$ reconstructed from its Taylor decomposition (Eq. 1) (dashed black). $\delta$pCO$_2$ calculated from monthly pCO$_2$ (solid yellow) is shown for comparison with the Taylor expansion. Column (a) shows the period 2006-2026 and column (b) shows the period 2080-2100. Each row represents the global zonal average for a different latitudinal band. Temperature dominates all latitudes except the Southern Ocean. In the 40°-60°N band, T contribution is largely compensated by DIC. The TA$_s$ and S$_{fw}$ effects are rather small in all latitudes.

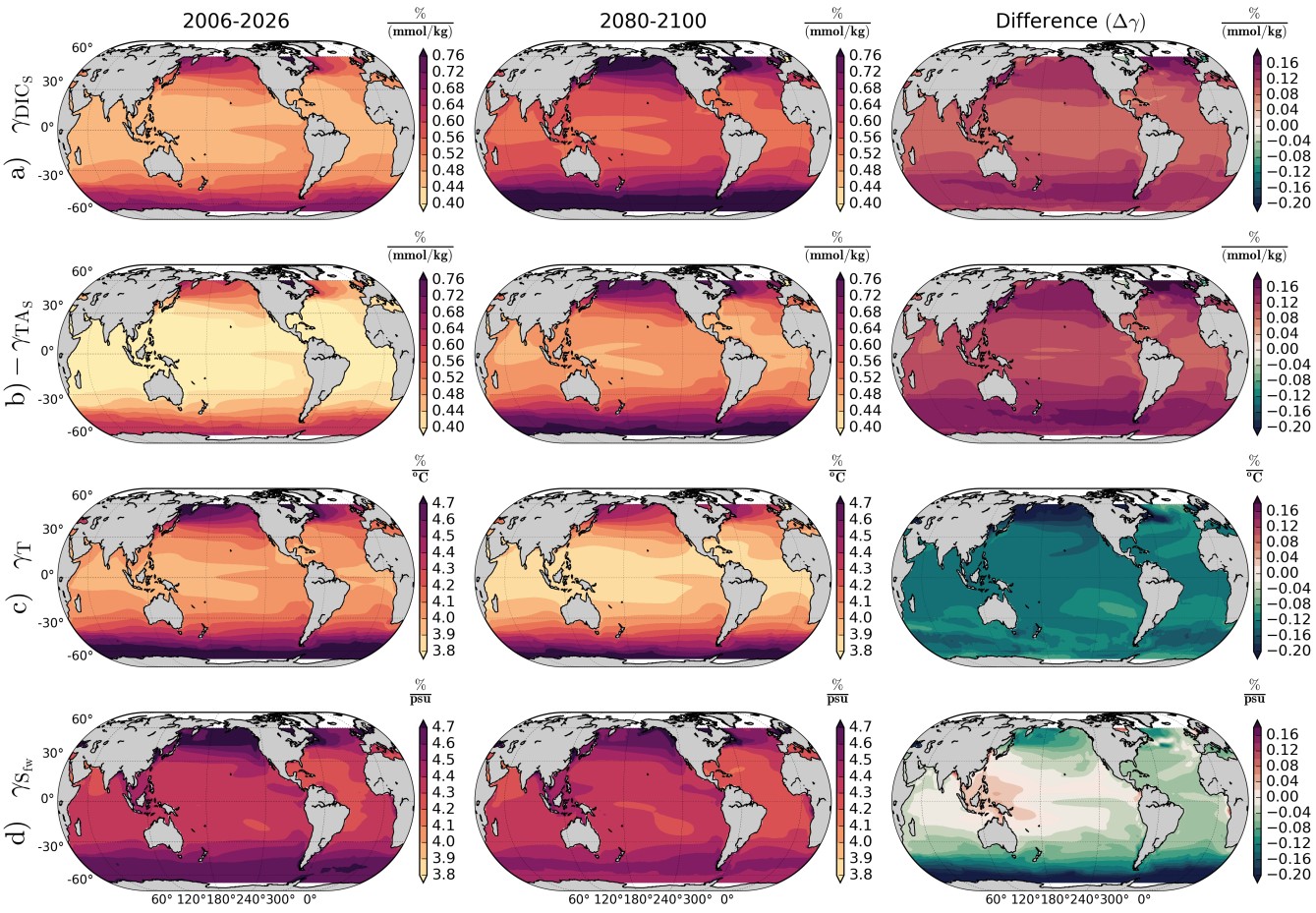

**Figure 4. RCP8.5 ensemble mean pCO$_2$ sensitivities:** for DIC$_s$ (row a), TA$_s$ (row b), T (row c) and S$_{fw}$ (row d). Row b) shows the negative of $\gamma_{TA}$. The first and second columns show the sensitivities by 2006-2026 and 2080-2100 respectively. The third column shows the difference between 2080-2100 and 2006-2026 sensitivities. High latitudes show the largest difference between initial and final periods. While DIC$_s$ and TA$_s$ sensitivities increase, the T and S$_f w$ sensitivities decreases, except in the Western Pacific Warm Pool, where $\gamma_{S_{fw}}$ increases.

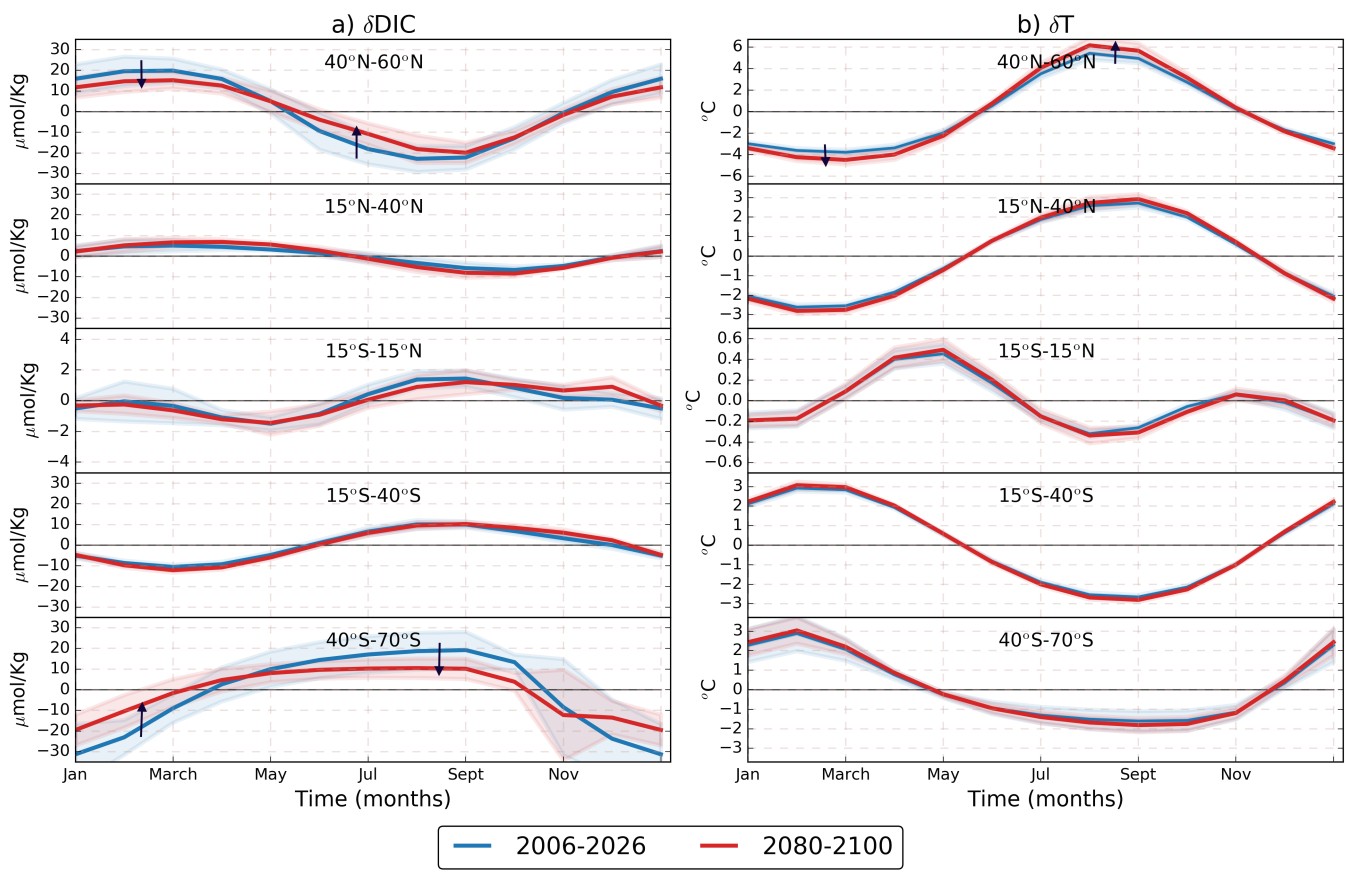

**Figure 5. RCP8.5 ensemble zonal mean seasonal cycles:** a) $\delta DIC_s$ and b) $\delta T$, for different latitudinal bands. Blue lines represent the 2006-2026 period, depicted for comparison with the 2080-2100 period shown by red lines. Different panels represent different latitudinal sections. Black arrows point out that while T seasonal cycle is projected to increase in most of the ocean, global $DIC_s$ is projected to decrease. The shading represents one standard deviation across the models. It is important to note that the scale is different for some of the latitudinal bands.

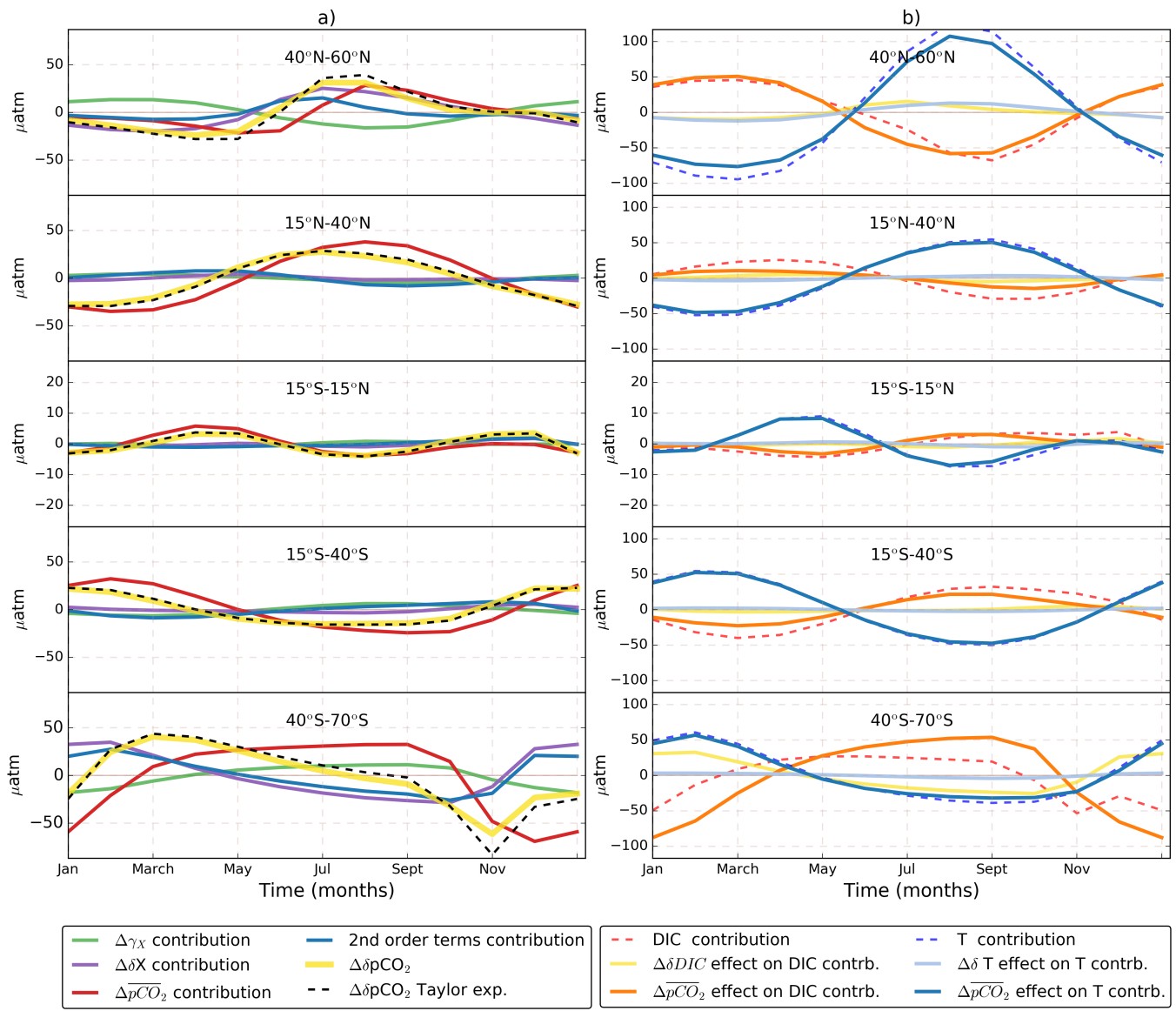

**Figure 6. Contribution of seasonalities, sensitivities, and mean pCO₂ changes to ΔδpCO₂.** a) Time series for the terms of Eq.(4) for different latitudinal bands. The $\Delta$ symbol represents the total century change, calculated as 2080-2100 value -minus- 2006-2026 value. The total change in seasonal pCO₂ ($\Delta\delta pCO_2$) is depicted as dashed black. This change is decomposed into changes in seasonalities ($\Delta\delta \boldsymbol{X}$, purple), sensitivities ($\Delta\gamma_X$, green), mean pCO₂ ($\Delta\overline{pCO_2}$, red) and second order terms (blue) summed over the four variables that control pCO₂ (DIC, TA, T and S). For comparison with the expansion, $\Delta\delta pCO_2$ is calculated from model output (yellow). Column b) shows the total change of DIC (dashed red) and T (dashed blue) contributions. Also shown, are two components of the total change on these contributions; the $\Delta\overline{pCO_2}$ effect on the DIC (solid orange) and T (solid blue) contributions, and the $\Delta\delta DIC$ (yellow) and $\Delta\delta T$ (light blue) effects. In column a), the $\delta pCO_2$ change follows the $\Delta\overline{pCO_2}$ effect. Column b) shows that actually, the leading cause of amplification is the $\Delta\overline{pCO_2}$ effect on the T contribution. It is important note the different scale between column a) and b)). Also, the scale was reduced in the 15°S-15°N region to highlight its features.

