# Peer review of "Drivers of future seasonal cycle changes of oceanic pCO2"

_Biogeosciences, 2018_

## Referee Comment (RC1) · Anonymous Referee #1 · 11 May 2018

Summary:

Gallego and co-authors investigate the magnitude and drivers of the seasonal changes in the sea surface pCO2. They use 7 CMIP5 models and compare the changing seasonality between 2 periods in the early 2000s and at the end of the coming century using the RCP8.5 scenario. The authors perform a Taylor expansion to investigate the relative contributions of the individual drivers (T, S, DIC, TA) as well as their changing sensitivities. The authors conclude that the seasonal pCO2 cycle will intensify by a factor of 1.5-3% by the end of the century, mainly owing to the sensitivity of the pCO2 cycle to changes in DIC and T, with both terms counteracting each other.

Strengths:

The changing seasonality in the surface ocean pCO2 and its potential impact on ocean acidification and marine life has recently received a lot of attention. More and more evidence emerges that the excess uptake of CO2 by the oceans will lead to environmental stress conditions, which will emerge earlier in time due to the seasonal pCO2 and pH amplification. The authors present here an extensive analysis building on state-of-the-art modelling output to estimate how strong the CO2 amplification is expected to be by the end of the century and what the main drivers of this amplification are. In my view, one strength of the conducted analysis is, that it nicely bridges between 2 recently published studies by Landschutzer et al 2018 and Kwiatkowski et al 2018 (both cited in the main text), hence I do believe the study has its place in the current literature and the results will be of interest to experts and the wider BG readership.

Weaknesses:

Unfortunately, while bridging between the current literature is the strong point of the presented manuscript, it also reveals its strongest weakness. On many occasions the authors fail to clearly highlight what is novel about their analysis and what has been previously shown. While the authors do give credit e.g. to the Landschutzer et al and Kwiatkowski et al studies at some place in the text (hence they must have read them), they fail to discuss their results in context to what is already known by these other studies. In some cases, the authors even create the impression that conclusions drawn here are novel, whereas they have been highlighted in other studies. To name the concrete examples:

.) Page 6 lines 1-2: "In general, towards the end of the century pCO2 amplifies more in high latitudes, . . ." this is the same result as for the past ears based on observational data (Landschutzer et al 2018, Figure 4) and for the future pH as a direct consequence of CO2) (Kwiatkowski et al 2018, Figure 3)

.) Page 9 lines 6-7: "We demonstrate that on average the global amplification of pCO2 is due to the overall longterm increase of anthropogenic CO2." This is the same conclusions Landschutzer et al 2018 reached based on examining trends in amplitude over the past 30 years, yet this is nowhere indicated. It is still a valuable result considering the focus of the study being the coming century, but it needs to be highlighted that other studies derive to the same conclusion.

.) Page 9 lines 11-12: "Our results extend and refine the current views, in which the future amplification has been attributed uniquely to the DIC sensitivity" – This is not correct. Both Landschutzer et al and Kwiatkowski et al discuss the attribution of other terms as well. The authors even briefly mention this in their introduction page 2 line 32: "Current literature suggests that the seasonal amplification is a consequence of an increase on the T and DIC contributions to pCO2 (Landschutzer et al 2018) ..."

.) Page 9 lines 17-19:"The first complete analytical Taylor expansion of pCO2 in terms of the variables DICs, TAs, T and S showed that DICs and T contributions are the main counteracting terms to control the pCO2, both under present-day and future conditions. The prevalence of one term over the other in various regions remains similar, even under enhanced CO2 conditions" – This has also been shown by Landschutzer et al 2018 under past/present conditions, yet again this is not mentioned anywhere. Furthermore, by stating "The first complete Tayler expansion . . ." I suppose the authors mean within their own study, yet it created the impression that the authors refer to the first complete Tayler expansion overall, whereas, e.g. Kwiatkowski et al use the same Tayler expansion in their analysis.

.) Page 9 lines 23-26: "Spatially, we found that the magnitude of the contributions depends on the mean pCO2 , its local sensitivities (DIC,TA,T,S) and the amplitude of their seasonal cycles ((DIC,TA,T,S)). The phases depend on the regional characteristics of the seasonal cycles and they moderate the counteracting nature of both contributions. The compensation of DICsÂǎ and T contributions is most effective when they are six months out of phase." This mirrors again a conclusion drawn in Landschutzer et al 2018 (see e.g. Figure 3 in their study), whereas a comparison, discussion or even mentioning of this circumstance is missing here. Also regional characteristic have been

discussed by Landschutzer et al 2018 and in terms of pH by Kwiatkowski et al 2018.

.) Another important result is only "hand wavy" introduced, namely that TA and S play a lesser role in the future pCO2 cycle amplification. One of the weak points of the Landschutzer et al 2018 study is that the authors ignore e.g. TA contributions, yet this study suggests that is of minor concern even when evaluating the century-long seasonal amplification. The authors also discuss second order terms here that have not been introduced in Landschutzer et al 2018 or Kwiatkowski et al 2018, but this is also not mentioned/compared.

.) Very interesting regional differences occur between the observation-based assessment of Landschutzer et al 2018 and this study, that are not discussed at all. Landschutzer et al find a DIC dominance in the high latitudes of both hemispheres, whereas the model based study suggests a T dominated increase in the high latitude northern hemisphere. Is this due to a model bias in seasonality. Is this the same across all models?

Recommendation:

The authors have conducted an extensive, interesting and certainly valuable analysis using state-of-the-art model outputs. Their methods are sound and their results nicely fit alongside the existing literature. The lack of discussion with the existing literature, however, is of major concern, particularly that the authors fail to acknowledge similar studies coming to the same conclusions. If the authors were to revise their manuscript and discuss their results in a fair way considering the existing literature, I believe this study can be considered for publication. The revisions however will affect the text throughout, hence I recommend major revisions of the manuscript.

Specific and minor comments to the text:

Abstract line 1: "observations" – its observation-based

Introduction page 1 line 22: a third of the anthropogenic CO2 produced by fossil fuel

burning, cement production and deforestation since the industrial revolution" – the cited Sabine study suggest 48% since the beginning of industrialization. The referenced 1/3 refer to the annual uptake as stated in the second study cited, namely the Le Quere et al carbon budget.

Page 2 line 21: [CO2(aq)]is introduced.ÂăFor the non carbonate seawater chemists that read BG it would be helpful to explain the difference between [CO2] and [CO2(aq)]

Page 4 line 11 and Supplement figure S1: The comparison between individual models gets worse in the high latitudes. Any idea why? The high latitude northern hemisphere is also wher this study differs from the observation-based analysis of Landschutzer et al 2018.

Page 4 line 20, equation 3 and following: the delta terms also represent the mean seasonal cycle over 20 years (period 1 or period 2) hence they should have also an overbar (like the pCO2) for consstency.

Page 5 line 14: "The range agrees with previous estimates by Takahashi et al. (2002)." Please add the comparison (visual or in table form), e.g. in the supplement for the readers of this study. Otherwise the reader has to jump around several different manuscripts for a simple comparison.

Page 5 line 21: "Our mean amplification factor estimation agrees with the lower end range of (McNeil et al. , 2016)." – Please add numbers for the reader of this study.

Page 6 lines 8-9: "Our estimated contributions from DICs and T to the present day pCO2 are in good agreement with the data based estimates (Takahashi et al. , 2002; Fay et al., 2017)." Please add a visual comparison or numbers for the readers of this study.

Page 7 lines 6-7: "DIC must not be confused with the Revelle factor, which is defined as R = DIC x gamma DIC." – this statement comes a bit out of the blue and while true it is not clear to me why it appears here. Based on the equations/wording used in this

study I don't see the danger that these terms are mixed up.

---

## Referee Comment (RC2) · Anonymous Referee #2 · 23 May 2018

**General comments:**

In this study, the authors assess future changes in the seasonal cycle of surface ocean  $pCO_2$  using simulations from 7 different CMIP5 Earth system models subjected to RCP8.5 forcing. A Taylor series decomposition approach is used to identify the important drivers of  $pCO_2$  seasonality and its future changes. The authors find that the  $pCO_2$  seasonal amplitude will increase by a factor of 1.5 to 3 by the end of the current century. The primary cause of this increase is the increase in ocean mean  $pCO_2$  (a response to increasing anthropogenic emissions), which enhances the  $pCO_2$  seasonal variation occurring in response to seasonal variations in temperature (T) and dissolved inorganic carbon (DIC). Changes in T and DIC seasonality at high latitudes are also relevant for understanding the model-simulated changes in  $pCO_2$  seasonality.

This is a nice study that complements some recent work (e.g., McNeil and Sasse, 2016; Landschützer et al., 2018; Kwiatkowski and Orr, 2018) examining the changing seasonality of ocean carbonate chemistry variables over recent decades and in the future. The paper is generally clear, well written and logically organized, and the scientific methods are sound. However, I do strongly agree with Referee #1's assessment that the authors should do a better job of placing their results in the context of previous work. While this is done to some extent already – and, in all fairness, the authors certainly cite the relevant literature – it tends to get a bit lost in the discussion and it's often a little unclear which results are novel and which simply confirm previous findings. It could be helpful to add a separate Discussion section before the Summary and Conclusions in which results from the current study are compared and contrasted with those from previous studies. In addition to this, I've included several specific comments and technical corrections below for the authors to consider. I feel that a suitably revised version of the manuscript – addressing the points raised here – should be publishable in Biogeosciences.

**Specific comments:**

1) p. 5, lines 19-20: "McNeil et al. (2016) using a data-based approach" – It would be good to clarify what you mean here by a "data-based approach".

2) p. 5, lines 23-25: "Using observations Landschützer et al. (2018) found..." – I have two issues with this sentence. First, it's unclear to me where the "mean 20 µatm increase by the end of the century" comes from; the values given earlier in this paragraph (see also Fig. 1) are significantly larger than this (e.g., 41 µatm increase between 40°S-40°N). Second, I wouldn't expect the rate of change of pCO2 seasonality in observations to match that in the CMIP5 models in the RCP8.5 simulations, since many of the important drivers of pCO2 variability (e.g., atmospheric CO2) are changing at much faster rates in the latter than they are in the former.

3) Fig. 3/Fig. S1: These figures show (among other things) that the Taylor expansion generally does a good job in reproducing the actual  $\delta pCO_2$  calculated from model output. However, there seems to be an inconsistency between the two figures. Specifically, Fig.

3 suggests that the Taylor expansion slightly overestimates the seasonal amplitude of  $pCO_2$  (this is most evident for the 40°S-70°S latitude band), while Fig. S1 suggests exactly the opposite: an underestimation of the seasonal amplitude.

4) p. 7, line 15: "decrease in the future to a global mean value of 0.035" – This number seems to be too small looking at Fig. 4c (middle column).

5) p. 7, line 26: "with lower temperatures in winter and higher in summer" – It might be good to clarify here that you do not mean lower temperatures in an absolute sense (i.e., winter temperatures are certainly projected to be higher at the end of century under RCP8.5 than they are at present).

6) p. 8, lines 26-27: "we decomposed the  $DIC_s$  and T contributions..." – I only see the seasonal cycle and mean pCO2 components in Fig. 6b, not the sensitivity component.

Technical corrections:

1) p. 1, line 2: Should be "a rate of 2-3 µatm per decade".

2) p. 3, line 5: "Methodology" misspelled.

3) p. 3, line 14: Should be "scarce".

4) p. 7, line 11: Should probably remove the word "change" here, since the annual cycle amplitude change is actually 168 μatm minus 96 μatm (i.e., 72 μatm).

5) p. 7, line 15: Should be "row (c)".

---

## Author Comment (AC1) · 3 Jul 2018

We thank the referee for reviewing the manuscript and for giving insightful and detailed comments that helped to improve our manuscript noticeably.

**Comment 1:**
**The changing seasonality in the surface ocean pCO$_2$ and its potential impact on ocean acidification and marine life has recently received a lot of attention. More and more evidence emerges that the excess uptake of CO$_2$ by the oceans will lead to environmental stress conditions, which will emerge earlier in time due to the seasonal pCO$_2$ and pH amplification. The authors present here an extensive analysis building on state-of-the-art modeling output to estimate how strong**

[Figure]

the $CO_2$ amplification is expected to be by the end of the century and what the main drivers of this amplification are. In my view, one strength of the conducted analysis is, that it nicely bridges between 2 recently published studies by *Landschützer et al.* (2018) and *Kwiatkowski and Orr* (2018) (both cited in the main text), hence I do believe the study has its place in the current literature and the results will be of interest to experts and the wider BG readership.

Unfortunately, while bridging between the current literature is the strong point of the presented manuscript, it also reveals its strongest weakness. On many occasions the authors fail to clearly highlight what is novel about their analysis and what has been previously shown. While the authors do give credit e.g. to the *Landschützer et al.* (2018) and *Kwiatkowski and Orr* (2018) studies at some place in the text (hence they must have read them), they fail to discuss their results in context to what is already known by these other studies. In some cases, the authors even create the impression that conclusions drawn here are novel, whereas they have been highlighted in other studies. To name the concrete examples:

**Response:** We revised large part of the manuscript to properly identify which findings are novel and which ones already exists in the current literature.

We added two supplementary figures: Fig. S3 that shows a comparison of $pCO_2$ seasonal amplitude by *Landschützer et al.* (2017) and *Takahashi et al.* (2014), as well as their thermal and non-thermal components. Fig. S4 shows a comparison of summer-minus-winter $pCO_2$ amplitude between models, for 2006-2026 and 2080-2100 periods. The figures are shown in the supplement of our response.

Below we address the specific referee's comments. Subsequently, we list other changes that were made to the manuscript, as well as references added.

**Comment 2: Page 6 lines 1-2: "In general, towards the end of the century $pCO_2$ amplifies more in high latitudes, . . .". This is the same result as for the past years based on observational data ((*Landschützer et al.*, 2018), Figure 4) and for the future pH as a direct consequence of $CO_2$ ((*Kwiatkowski and Orr*, 2018),**

**Figure 3)).**

**Response:** We changed the sentence to: " In general, towards the end of the century the $pCO_2$ amplifies more in high latitudes, but so does the standard deviation uncertainty among models. This regional pattern agrees with the observation-based findings of *Landschützer et al.* (2018) which show that high latitudes have already experienced a larger amplification than mid-low latitudes from 1982 to 2015. Furthermore, the same pattern is projected by CMIP5 models for the seasonal amplification of $[H^+]$ by the end of the century (*Kwiatkowski and Orr*, 2018). This is expected from the near-linear relation between $pCO_2$ and $[H^+]$."

**Comment 3: Page 9 lines 6-7: "We demonstrate that on average the global amplification of $pCO_2$ is due to the overall longterm increase of anthropogenic $CO_2$". This is the same conclusions *Landschützer et al.* (2018) reached based on examining trends in amplitude over the past 30 years, yet this is nowhere indicated. It is still a valuable result considering the focus of the study being the coming century, but it needs to be highlighted that other studies derive to the same conclusion.**

**Response:** We changed the sentence to: "The projected amplification by the earth-system models and the possible causes of it, are consistent with observation-based amplification for the period from 1982 to 2015 (*Landschützer et al.*, 2018). In agreement with the observational results, also the model projections towards the end of this century demonstrate that the global amplification of $\delta pCO_2$ is due to the overall longterm increase of anthropogenic $CO_2$. A higher oceanic $CO_2$ concentration enhances the effect of solubility changes on $\delta pCO_2$ and alters the seawater carbonate chemistry, also enhancing the DIC seasonality effect."

**Comment 4: Page 9 lines 11-12: "Our results extend and refine the current views, in which the future amplification has been attributed uniquely to the DIC sensitivity". This is not correct. Both *Landschützer et al.* (2018)and *Kwiatkowski and Orr* (2018) discuss the attribution of other terms as well. The authors even briefly mention this in their introduction page 2 line 32: "Current literature suggests that the seasonal amplification is a consequence of an increase on the T and DIC contributions to pCO2 (*Landschützer et al.*, 2018)..."**
**Response:** We agree, we removed the sentence.

**Comment 5: Page 9 lines 17-19: "The first complete analytical Taylor expansion of $pCO_2$ in terms of the variables DICs, TAs, T and S showed that DICs and T contributions are the main counteracting terms to control the $pCO_2$, both under present-day and future conditions. The prevalence of one term over the other in various regions remains similar, even under enhanced $CO_2$ conditions". This has also been shown by *Landschützer et al.* (2018) under past/present conditions, yet again this is not mentioned anywhere. Furthermore, by stating "The first complete Taylor expansion . . ." I suppose the authors mean within their own study, yet it created the impression that the authors refer to the first complete Taylor expansion overall, whereas, e.g. *Kwiatkowski and Orr* (2018) use the same Taylor expansion in their analysis.**
**Response:** By "first complete analytical Taylor expansion, "we refer to the incorporation of T and S analytical terms, and therefore it is the first complete with analytical expressions in the four terms, which - to our knowledge- has not been done before. However, we agree this might be misleading, so we changed the sentence to: "The models confirm the well-established mechanisms controlling present-day $\delta pCO_2$ (*Takahashi et al.*, 2002; *Sarmiento and Gruber*, 2006; *Fay and McKinley*, 2017). $DIC_s$ and T contributions are the main counteracting terms dominating the seasonal evolution of $\delta pCO_2$. Furthermore, the models show that under future conditions

the controlling mechanisms remain unchanged. This result confirms the findings of *Landschützer et al.* (2018) that identified the same regional controlling mechanism for the past 30 years. The relative role of the DIC and T terms is regionally dependent. High latitudes and upwelling regions, such as the California Current system and the coast of Chile, are dominated by $DIC_s$ and the temperate low latitudes are driven by T. Only in the North Atlantic and North-Western Pacific the models show a dominance of thermal effects over non-thermal effects, which is in disagreement with observations. This further illustrates the urgent need for models to accurately represent regional oceanographic features. " The discussion on the difference between models and observations was added in the results and discussion section.

**Comment 6: Page 9 lines 23-26: "Spatially, we found that the magnitude of the contributions depends on the mean $pCO_2$ , its local sensitivities (DIC,TA,T,S) and the amplitude of their seasonal cycles ((DIC,TA,T,S)). The phases depend on the regional characteristics of the seasonal cycles and they moderate the counteracting nature of both contributions. The compensation of DICs and T contributions is most effective when they are six months out of phase." This mirrors again a conclusion drawn in *Landschützer et al.* (2018) (see e.g. Figure 3 in their study), whereas a comparison, discussion or even mentioning of this circumstance is missing here. Also regional characteristic have been discussed by *Landschützer et al.* (2018) and in terms of pH by *Kwiatkowski and Orr* (2018) .**
**Response:** The sentence was changed to: "Moreover, the $pCO_2$ seasonal cycle amplitude depends on the relative magnitude and phase of the contributions. Spatially, we found that the magnitude of the contributions depends on the mean $pCO_2$, its local sensitivities ($\gamma_{DIC,TA,T,S}$) and the amplitude of their seasonal cycles ($\delta(DIC,TA,T,S)$). The phases depend on the regional characteristics of the seasonal cycles and they moderate the counteracting nature of both contributions. The ensemble mean reproduces the highly effective compensation of $DIC_s$ and T contributions

when they are six months out of phase, confirming previous studies (*Takahashi et al.*, 2002; *Landschützer et al.*, 2018)."

**Comment 7: Another important result is only "hand wavy" introduced, namely that TA and S play a lesser role in the future pCO2 cycle amplification. One of the weak points of the *Landschützer et al.* (2018) study is that the authors ignore e.g. TA contributions, yet this study suggests that is of minor concern even when evaluating the century-long seasonal amplification. The authors also discuss second order terms here that have not been introduced in *Landschützer et al.* (2018) or *Kwiatkowski and Orr* (2018) , but this is also not mentioned/compared.**
**Response:** We added in the conclusions, page 9, line 21: "The TA and S terms have a small impact in most regions, except on the high latitudes where the TA contribution complements the DIC one, enhancing the non-thermal effect in this region. Interestingly, in the high latitudes, the amplification through second order terms are as important as the change in the seasonality of the drivers. Their high values arise from changes in mean $pCO_2$ acting over the changing T seasonality. "

**Comment 8: Very interesting regional differences occur between the observation-based assessment of *Landschützer et al.* (2018) and this study, that are not discussed at all. *Landschützer et al.* (2018) find a DIC dominance in the high latitudes of both hemispheres, whereas the model based study suggests a T dominated increase in the high latitude northern hemisphere. Is this due to a model bias in seasonality. Is this the same across all models?.**
**Response:** We added in the "Results and discussion" section, page 6, line 12: "The models show that the $\delta pCO_2$ in the 40°N to 60°N band is controlled by T, which disagrees with the above mentioned observations that show a non-temperature dominance in this band. The difference between models and observations arises from two regions: the North Atlantic basin and the North Western Pacific; specifically near

the Oyashio Current, and the outflows from the Okhotsk Seas (see Supplementary Fig. S3). Most models show a T dominance in the North Atlantic basin; only CESM1-BGC and GFDL-ESM2M show a DIC dominance (see Supplementary Fig. S4). The North Atlantic is one of the major sinks of anthropogenic $CO_2$, however some models fail to estimate its uptake capacity (*Goris et al.*, 2018). *Goris et al.* (2018) found that models with an efficient carbon sequestration present a DIC-dominated $pCO_2$ seasonal cycle in the North Atlantic, but models with low anthropogenic uptake show a T dominance in this region. In the North-Western Pacific, *Mckinley et al.* (2006) found that coarse models are not able to capture the intricate oceanographic features of this area, and therefore the $pCO_2$ seasonality is not well captured."

**Comment 9: The authors have conducted an extensive, interesting and certainly valuable analysis using state-of-the-art model outputs. Their methods are sound and their results nicely fit alongside the existing literature. The lack of discussion with the existing literature, however, is of major concern, particularly that the authors fail to acknowledge similar studies coming to the same conclusions. If the authors were to revise their manuscript and discuss their results in a fair way considering the existing literature, I believe this study can be considered for publication. The revisions however will affect the text throughout, hence I recommend major revisions of the manuscript.**
**Response:** As suggested by the referee, we have done a major revision of the manuscript. We thank again for the suggestions that helped to improve the manuscript; we placed our results and their relevance among the current literature and compared/contrasted our findings which previous results, in particular those by *Landschützer et al.* (2018).

**Comment 10: Abstract line 1: "observations" its observation-based**
**Response:** Changed to "observation-based results"

**Comment 11: Introduction page 1 line 22: a third of the anthropogenic CO$_2$ produced by fossil fuel burning, cement production and deforestation since the industrial revolution". The cited Sabine study suggest 48% since the beginning of industrialization. The referenced 1/3 refer to the annual uptake as stated in the second study cited, namely the Le Quere et al carbon budget.**
**Response:** We changed it to: "the ocean has absorbed nearly half of the anthropogenic CO$_2$ produced by fossil fuel burning and cement production since the industrial revolution (*Sabine et al.*, 2004)"

**Comment 12: Page 2 line 21: [CO2(aq)]is introduced. For the non carbonate seawater chemists that read BG it would be helpful to explain the difference between [CO2] and [CO2(aq)]**
**Response:** We changed it to: " This is due to the ability of CO$_2$ to react with seawater to form bicarbonate [HCO$_3^-$] and carbonate [CO$_3^{2-}$], leaving only a small portion of the dissolved carbon dioxide in the form of aqueous CO$_2$ ([CO$_2$(aq)]). [CO$_2$(aq)] together with the carbonic acid ([H$_2$CO$_3$]) are defined as [CO$_2$]. Therefore, it is useful to define the total amount of carbon as DIC, which is the sum of the three carbon species ([HCO$_3^-$], [CO$_3^{2-}$] and [CO$_2$])."

**Comment 13: Page 4 line 11 and Supplement figure S1: The comparison between individual models gets worse in the high latitudes. Any idea why? The high latitude northern hemisphere is also where this study differs from the**

**observation-based analysis of *Landschützer et al.* (2018).**
**Response:** In this figure we compare the $pCO_2$ amplification calculated as model output with the value from the Taylor expansion. The Taylor expansion is less precise in higher latitudes, probably because second order terms gain importance. The difference with *Landschützer et al.* (2018) was addressed in comment 8.

**Comment 14: Page 4 line 20, equation 3 and following: the delta terms also represent the mean seasonal cycle over 20 years (period 1 or period 2) hence they should have also an overbar (like the pCO2) for consistency.**
**Response:** We leave the nomenclature as it is, as by "mean" we refer to the mean value of the data, instead of the deviation of the mean, which is the seasonal cycle.

**Comment 15: Page 5 line 14: "The range agrees with previous estimates by *Takahashi et al.* (2002)." Please add the comparison (visual or in table form), e.g. in the supplement for the readers of this study. Otherwise the reader has to jump around several different manuscripts for a simple comparison.**
**Response:** We added a supplementary figure S3, for better comparison with data from *Takahashi et al.* (2014), for a reference year 2005 and with *Landschützer et al.* (2017). We also added at page 5, after line 14 : "The ensemble mean initial seasonal amplitude range is in good agreement with observational estimates calculated for the reference year 2005 (*Takahashi et al.*, 2014), and for the 1982-2015 period (*Landschützer et al.*, 2017). The agreement between models and observations is remarkably good in the equatorial regions, but the initial amplitude is slightly overestimated in the mid and high latitudes (see Supplementary Fig. S3).The higher amplitude in models than observations is expected, as the initial period 2006-2026 already experienced an amplification compared to previous years. Moreover, *Tjiputra et al.* (2014) found

that the ocean's $pCO_2$ historical trend is larger in models than observations when it is estimated in large scale areas of the ocean. However, they found that models' $pCO_2$ trends agree with observations when the trends are subsampled to the locations where the observations were taken, and therefore they do a good job reproducing well-known time series. Moreover, differences are expected as *Pilcher et al.* (2015) suggested that CMIP5 models perform well in reproducing the seasonal cycle timing, but still show considerable errors in reproducing the seasonal amplitude of $pCO_2$ due to differences in the mechanisms represented in each model, especially in subpolar biomes. "

**Comment 16: Page 5 line 21: "Our mean amplification factor estimation agrees with the lower end range of *McNeil and Sasse* (2016)." Please add numbers for the reader of this study.**
**Response:** We changed this sentence to: " Our mean amplification factor estimation agrees with the threefold amplification found for most of the ocean by *McNeil and Sasse* (2016)."

**Comment 17: Page 6 lines 8-9: "Our estimated contributions from DICs and T to the present day $pCO_2$ are in good agreement with the data based estimates (*Takahashi et al.*, 2002; *Fay and McKinley*, 2017)." Please add a visual comparison or numbers for the readers of this study.**
**Response:** Instead of comparing with *Takahashi et al.* (2002), and *Fay and McKinley* (2017), we used the dataset of *Takahashi et al.* (2014) and calculated thermal and non thermal components for year 2005. We also added a comparison with the thermal and non-thermal components for years 1982-2015 that Peter Landschützer kindly provided to us. The results are shown in Supplementary Figure S3, and the discussion

was added in section 3.2: "For most of the ocean, the ensemble mean estimated contributions from $DIC_s$ and T to the present-day $\delta pCO_2$ are in good agreement with the data-based estimates of *Takahashi et al.* (2014); *Landschützer et al.* (2017), particularly in the equatorial regions (see Supplementary Fig. S3). However our temperature and DIC contributions are slightly larger in mid and high latitudes, for the same reasons the $pCO_2$ seasonal amplitude is overestimated (see Section 3.1). Also, differences arise between our $DIC_s$ contribution and the observation-based so called "non-thermal" contribution, because the non-thermal contribution also includes the total alkalinity and salinity effects. Nonetheless, between 40°S-40°N our ensemble mean shows that $\delta pCO_2$ is dominated by changes in temperature that control $CO_2$ solubility, which decreases in summer enhancing $pCO_2$, in agreement with observations. The Southern Ocean is controlled by DIC, that responds to changes in upwelling and phytoplankton blooms. Both mechanisms act together to decrease (increase) DIC in summer (winter) (*Sarmiento and Gruber*, 2006). " The discussion of northern high latitudes is added in comment 8.

**Comment 18: Page 7 lines 6-7: "DIC must not be confused with the Revelle factor, which is defined as R = DIC x gamma DIC". This statement comes a bit out of the blue and while true it is not clear to me why it appears here. Based on the equations/wording used in this study I don't see the danger that these terms are mixed up.**
**Response:** The Revelle factor and the sensitivity are different, and sometimes confused. We included the relationship because *Takahashi et al.* (1993) computed the Revelle factor. This sentence was rearranged as: "This follows the approach of *Takahashi et al.* (1993), however instead of computing the Revelle factor we use $\gamma_{DIC}$, both terms are related by $R = DIC \cdot \gamma_{DIC}$."

**Other changes added:**

- page 6, line 26 we added: " In this region some models underestimate the $pCO_2$ trend (*Tjiputra et al.*, 2014), and therefore the seasonal amplification might be underestimated too."

- page 7, line 1, we added: "Lower buffer factors (higher sensitivities factors) are found in regions where DIC and TA have similar values, and they will decrease (increase) as the DIC/TA ratio in the oceans increases (*Egleston et al.*, 2010). "

- page 7, line 7, we added: "$\gamma_{S_{fw}}$ decreases everywhere except in the Western Pacific Warm Pool. In this region $\gamma_{S_{fw}}$ increases probably due future changes in precipitation that enhance the fresh-water effect."

- page 7, line 33 was changed to: "*Kwiatkowski and Orr* (2018) demonstrated that the seasonality of the drivers is important to determine future changes in $[H^+]$ seasonality. In the same fashion, our results show that the four $\delta pCO_2$ drivers present changes in seasonality, and in particular $\delta DIC_s$ and $\delta T$ changes are important to explain future projections of the $\delta pCO_2$ amplitude."

**References**

Egleston, E. S., Sabine, C. L., and Morel, F. M. M.: Revelle revisited: Buffer factors that quantify the response of ocean chemistry to changes in DIC and alkalinity, Global Biogeochem. Cycles, 24, GB1002, 2010.

Fay, A. R. and McKinley, G. A.: Correlations of surface ocean $pCO_2$ to satellite chlorophyll on monthly to interannual timescales, Global Biogeochem. Cycles, 31, 436-455, 2017.

Goris, N., Tjiputra, J., Olsen, A., Schwinger, J., Lauvset, S. K., and Jeansson, E.: Constraining projection-based estimates of the future North Atlantic carbon uptake, Journal of Climate, 31(10), 3959-3978, 2018.

Kwiatkowski, L. and Orr, J.: Diverging seasonal extremes for ocean acidification during the twenty-first century, Nat. Clim. Change, 8, 141-145, 2018.

Landschützer, P., Gruber, N., and Bakker, D.: An updated observation- based global monthly gridded sea surface $pCO_2$ and air-sea $CO_2$ flux product from 1982 through 2015 and its monthly climatology (NCEI Accession 0160558). Version 2.2. NOAA National Centers for Environmental Information. Dataset. doi:10.7289/V5Z899N6, 2017.

Landschützer, P., Gruber, N., Bakker, D. C. E., Stemmler, I., and Six, K. D.: Strengthening seasonal marine $CO_2$ variations due to increasing atmospheric $CO_2$, Nat. Clim. Change, 8, 146-150, 2018.

Mckinley, G. A., Takahashi, T., Buitenhuis, E., Chai, F., Christian, J. R., Doney, S. C., Jiang, M., Lindsay, K., Moore, J. K., Quere, C. L., Lima, I., Murtugudde, R., Shi, L., and Wetzel, P.: North Pacific carbon cycle response to climate variability on seasonal to decadal timescales, J. Geophys. Res., 111, C07S06, 2006.

McNeil, B. I. and Sasse, T. P.: Future ocean hypercapnia driven by anthropogenic amplification of the natural $CO_2$ cycle, Nature, 529, 383-386, 2016.

Pilcher, D. J., Brody, S. R., Johnson, L., and Bronselaer, B.: Assessing the abilities of CMIP5 models to represent the seasonal cycle of surface ocean $pCO_2$, J. Geophys. Res. Oceans, 120, 4625-4637, 2015.

Sabine, C. L., Feely, R. A., Gruber, N., Key, R. M., Lee, K., Bullister, J. L., Wanninkhof, R., Wong, C., Wallace, D. W. R., Tilbrook, B., Millero, F. J., Peng, T. H., Kozyr, A., Ono, T., and Rios, A. F.: The oceanic sink for anthropogenic $CO_2$, Science, 305, 367-371, 2004.

Sarmiento, J. L. and Gruber, N.: Ocean Biogeochemical Dynamics, Princeton University Press, Princeton, New Jersey, USA, 2006.

Takahashi, T., Olafsson, J., Goddard, J. G., Chipman, D. W., and Sutherland, S. C.: Seasonal variation of $CO_2$ and nutrients in the high-latitude surface oceans: A comparative study, Global Biogeochem. Cycles, 7, 843-878, 1993.

Takahashi, T., Sutherland, S. C., Sweeney, C., Poisson, A., Metzl, N., Tilbrook, B., Bates, N., Wanninkhof, R., Feely, R. A., Sabine, C. L., Olafsson, J., and Nojiri, Y.: Global sea-air $CO_2$ flux based on climatological surface ocean $pCO_2$, and seasonal biological and temperature effects, Deep-Sea Research II, 49, 1601–1623, 2002.

Takahashi, T., Sutherland, S. C., Chipman, D. W., Goddard, J. G., New- berger, T., and Sweeney, C.: Climatological Distributions of pH, $pCO_2$, Total $CO_2$, Alkalinity, and $CaCO_3$ Saturation in the Global Surface Ocean. ORNL/CDIAC-160, NDP-094. Carbon Dioxide Information Analysis Center, https://doi.org/10.3334/CDIAC/OTG.NDP094, 2014.

Tjiputra, J. F., Olsen, A. R. E., Bopp, L., Lenton, A., Pfeil, B., Roy, T., Segschneider, J., Totterdell, I. A. N., and Heinze, C.: Long-term surface $pCO_2$ trends from observations and models, Tellus B, 66, 23 083, 2014.

**Supplement:**

[Figure]

Figure S3: Column a) shows the pCO$_2$ seasonal amplitude calculated as summer minus winter for each hemisphere respectively. b) and c) show the thermal and non-thermal contributions to pCO$_2$ seasonality respectively. First row shows CMIP5 models ensemble mean for the 2006-2026 period under the RCP8.5 scenario. Second row shows the estimates from Takahashi et al., (2014b) dataset for a reference year 2005, with summer-minus-winter thermal and non-thermal contributions calculated as Takahashi et al., (2002). Third rows show the same components for the Landschützer et al., (2017) pCO$_2$ data-set, and the thermal and non-thermal estimations that Peter Landschützer facilitated us, for the period 1982-2015.

[Figure]

Figure S4: $\delta$pCO$_2$ climatology for column a) 2006-2026 and b) 2080-2100 periods calculated as in Fig. S2; each row is the result for a different model. column c) shows the differences between column b) and a).

---

## Author Comment (AC2) · 3 Jul 2018

We are thankful for the referee's comments. The referee's specific comments helped with the revision of our calculations and improved the manuscript.

**Comment 1: In this study, the authors assess future changes in the seasonal cycle of surface ocean pCO$_2$ using simulations from 7 different CMIP5 Earth system models subjected to RCP8.5 forcing. A Taylor series decomposition approach is used to identify the important drivers of pCO$_2$ seasonality and its future changes. The authors find that the pCO$_2$ seasonal amplitude will increase by a factor of 1.5 to 3 by the end of the current century. The primary cause of this increase is the increase in ocean mean pCO$_2$ (a response to increasing anthropogenic emissions), which enhances the pCO$_2$ seasonal variation occurring**

[Figure]

in response to seasonal variations in temperature (T) and dissolved inorganic carbon (DIC). Changes in T and DIC seasonality at high latitudes are also relevant for understanding the model-simulated changes in pCO$_2$ seasonality. This is a nice study that complements some recent work (e.g., *McNeil and Sasse* (2016); *Landschützer et al.* (2018); *Kwiatkowski and Orr* (2018)) examining the changing seasonality of ocean carbonate chemistry variables over recent decades and in the future. The paper is generally clear, well written and logically organized, and the scientific methods are sound. However, I do strongly agree with Referee 1's assessment that the authors should do a better job of placing their results in the context of previous work. While this is done to some extent already and, in all fairness, the authors certainly cite the relevant literature it tends to get a bit lost in the discussion and it's often a little unclear which results are novel and which simply confirm previous findings. It could be helpful to add a separate Discussion section before the Summary and Conclusions in which results from the current study are compared and contrasted with those from previous studies. In addition to this, I've included several specific comments and technical corrections below for the authors to consider. I feel that a suitably revised version of the manuscript - addressing the points raised here - should be publishable in Biogeosciences.

**Response:** We revised the "Results and discussion" section, where we compared and contrasted our results with the existing literature. We added several changes in this section based on referee 1 and 2 comments. Most of the changes were discussed in the response to referee's 1 comments. In what follows we address referee 2's specific comments:

**Comment 2:** p. 5, lines 19-20: "*McNeil and Sasse* (2016) using a data-based approach" It would be good to clarify what you mean here by a "data-based approach".

**Response:** We changed to "*McNeil and Sasse* (2016) used observations and a neural-network-clustering algorithm to project that by year 2100..."

**Comment 3: p. 5, lines 23-25: "Using observations *Landschützer et al.* (2018) found..." I have two issues with this sentence. First, it's unclear to me where the mean 20 muatm increase by the end of the century comes from; the values given earlier in this paragraph (see also Fig. 1) are significantly larger than this (e.g., 41 muatm increase between 40°S-40°N). Second, I wouldn't expect the rate of change of $pCO_2$ seasonality in observations to match that in the CMIP5 models in the RCP8.5 simulations, since many of the important drivers of $pCO_2$ variability (e.g., atmospheric $CO_2$) are changing at much faster rates in the latter than they are in the former.**

**Response:** We did three changes based on this comment: We corrected the calculation and the sentence was changed to: " Using observations, *Landschützer et al.* (2018) found an increase of 2.2 $\mu$atm per decade, which is smaller than our findings of a total 42 $\mu$atm increase by the end of the century between 40°S-40°N, and a global-mean change of 81 $\mu$atm on the high latitudes. This difference is again possibly due the higher mean $pCO_2$ values in models than observations. " The discussion of the difference between models and observations was addressed in the response to Referee 1, Comment 15.

**Comment 4: Fig. 3/Fig. S1: These figures show (among other things) that the Taylor expansion generally does a good job in reproducing the actual $pCO_2$ calculated from model output. However, there seems to be an inconsistency between the two figures. Specifically, Fig.3 suggests that the Taylor expansion slightly overestimates the seasonal amplitude of $pCO_2$ (this is most evident for the 40°S-70°S latitude band), while Fig. S1 suggests exactly the opposite: an underestimation of the seasonal amplitude.**

**Response:** The labels of the S1 figure's axis were switched (x-axis label was y-axis, and vice versa). Figure S1 was corrected.

**Comment 5: p. 7, line 15: "decrease in the future to a global mean value of 0.035" .This number seems to be too small looking at Fig. 4c (middle column).**
**Response:** There was an error in the calculation. The sentence was changed to: "This value agrees with our global mean ensemble estimate of 0.0428. However, our analytical expression of $\gamma_T$ shows that this value varies regionally and, by reasons unknown to us, it might decrease in the future to a global mean value of 0.0415, (Fig. 4, row (c), third column). "

**Comment 6: p. 7, line 26: "with lower temperatures in winter and higher in summer" It might be good to clarify here that you do not mean lower temperatures in an absolute sense (i.e., winter temperatures are certainly projected to be higher at the end of century under RCP8.5 than they are at present).**
**Response:** This sentence was changed and we added some context: "All models show a slight increase in $\delta$T, only one model showed a slightly decrease in the southern region, and two models showed a decrease in the equatorial region during October to December. It is important to note that Fig. 5 shows the seasonal values, with the mean T removed. Therefore, when considering the positive T trends, the absolute summer values show an increase and the winter values a decrease. This agrees with the results of *Alexander et al.* (2018); who showed that models project a seasonal intensification of T, with larger warm extremes and reduced cold extremes. The authors attributed the T seasonality intensification to an increased oceanic stratification and an overall shoaling of the mixed layer depth, which confines seasonal changes in a reduced volume of water, producing larger changes at the surface. They

also showed that the intensification trends are stronger in summer than winter, as the mixed layer depth is shallower in summer. Moreover, ice covered regions will experience the largest increase in T seasonality, as the ice melting/freezing modulates the surface water temperature (*Carton et al.*, 2015). "

**Comment 7: p. 8, lines 26-27: "we decomposed the DICs and T contributions..." I only see the seasonal cycle and mean pCO$_2$ components in Fig. 6b, not the sensitivity component.**
**Response:** We changed the sentence to: "To further disentangle which of the two main drivers (DIC$_s$ or T) is most affected by $\Delta\overline{pCO_2}$, we decomposed the DIC$_s$ and T contributions in their sensitivity, seasonal cycle and $\overline{pCO_2}$ components. Figure 6, (b), shows the total DIC and T components and the $\Delta\overline{pCO_2}$ and seasonal cycles effects. The effects from the sensitivities are not depicted, as they only play a minor role. Only the $\Delta\gamma_{DIC}$ term gains importance in the Southern Ocean (not shown). "

**Comment 8: Technical corrections: 1) p. 1, line 2: Should be a rate of 2-3 muatm per decade?.**
Changed to decade.
**2) p. 3, line 5: "Methodology" misspelled.**
Corrected.
**3) p. 3, line 14: Should be scarce?.**
Changed to "scarced".
**4) p. 7, line 11: Should probably remove the word "change" here, since the annual cycle amplitude change is actually 168 muatm minus 96 muatm (i.e., 72 muatm).**
We removed the word "change" and changed the sentence to: "therefore, for a $\overline{pCO_2}$

equal to 800 $\mu$atm, the $\delta$pCO$_2$ amplitude due to $\delta$DIC amounts to 168 $\mu$atm. "
**5) p. 7, line 15: Should be "row (c)".**
Changed to "row c)"

**References**

Alexander, M. A., Scott, J. D., Friedland, K. D., Mills, K. E., Nye, J., Pershing, A. J., and Thomas, A. C.: Projected sea surface temperatures over the 21st century: Changes in the mean, variability and extremes for large marine ecosystem regions of Northern Oceans, Elem. Sci. Anth., 6, 9, 2018.

Carton, J. A., Ding, Y., and Arrigo, K. R.: The seasonal cycle of the Arctic Ocean under climate change, Geophys. Res. Lett., 42, 7681-7686, 2015.

Kwiatkowski, L. and Orr, J.: Diverging seasonal extremes for ocean acidification during the twenty-first century, Nat. Clim. Change, 8, 141-145, 2018.

Landschützer, P., Gruber, N., Bakker, D. C. E., Stemmler, I., and Six, K. D.: Strengthening seasonal marine CO$_2$ variations due to increasing atmospheric CO$_2$, Nat. Clim. Change, 8, 146-150, 2018.

McNeil, B. I. and Sasse, T. P.: Future ocean hypercapnia driven by anthropogenic amplification of the natural CO$_2$ cycle, Nature, 529, 383-386, 2016.

---

## Author Response (AR1)

**Response to Referee 1**

M. Angeles Gallego, Axel Timmermann, Tobias Friedrich, Richard E. Zeebe

July 17, 2018

We thank the referee for reviewing the manuscript and for giving insightful and detailed comments that helped to improve our manuscript noticeably.

**Comment 1:**

**The changing seasonality in the surface ocean $pCO_2$ and its potential impact on ocean acidification and marine life has recently received a lot of attention. More and more evidence emerges that the excess uptake of $CO_2$ by the oceans will lead to environmental stress conditions, which will emerge earlier in time due to the seasonal $pCO_2$ and pH amplification. The authors present here an extensive analysis building on state-of-the- art modeling output to estimate how strong the $CO_2$ amplification is expected to be by the end of the century and what the main drivers of this amplification are. In my view, one strength of the conducted analysis is, that it nicely bridges between 2 recently published studies by Landschützer et al. (2018) and Kwiatkowski and Orr (2018) (both cited in the main text), hence I do believe the study has its place in the current literature and the results will be of interest to experts and the wider BG readership.**

**Unfortunately, while bridging between the current literature is the strong point of the presented manuscript, it also reveals its strongest weakness. On many occasions the authors fail to clearly highlight what is novel about their analysis and what has been previously shown. While the authors do give credit e.g. to the Landschützer et al. (2018) and Kwiatkowski and Orr (2018) studies at some place in the text (hence they must have read them), they fail to discuss their results in context to what is already known by these other studies. In some cases, the authors even create the impression that conclusions drawn here are novel, whereas they have been highlighted in other studies. To name the concrete examples:**

**Response:** We revised large part of the manuscript to properly identify which findings are novel and which ones already exists in the current literature.

We added two supplementary figures: Fig. S3 that shows a comparison of

pCO$_2$ seasonal amplitude by Landschützer et al. (2017) and Takahashi et al. (2014), as well as their thermal and non-thermal components. Fig. S4 shows a comparison of summer-minus-winter pCO$_2$ amplitude between models, for 2006-2026 and 2080-2100 periods.

Below we address the specific referee's comments. Subsequently, we list other changes that were made to the manuscript, as well as references added.

**Comment 2: Page 6 lines 1-2: "In general, towards the end of the century pCO$_2$ amplifies more in high latitudes, . . .". This is the same result as for the past years based on observational data ((Landschützer et al., 2018), Figure 4) and for the future pH as a direct consequence of CO$_2$ ((Kwiatkowski and Orr, 2018), Figure 3)).**

**Response:** We changed the sentence to: " In general, towards the end of the century the pCO$_2$ amplifies more in high latitudes, but so does the standard deviation uncertainty among models. This regional pattern agrees with the observation-based findings of Landschützer et al. (2018) which show that high latitudes have already experienced a larger amplification than mid-low latitudes from 1982 to 2015. Furthermore, the same pattern is projected by CMIP5 models for the seasonal amplification of [H$^+$] by the end of the century (Kwiatkowski and Orr, 2018). This is expected from the near-linear relation between pCO$_2$ and [H$^+$]."

**Comment 3: Page 9 lines 6-7: "We demonstrate that on average the global amplification of pCO$_2$ is due to the overall longterm increase of anthropogenic CO$_2$". This is the same conclusions Landschützer et al. (2018) reached based on examining trends in amplitude over the past 30 years, yet this is nowhere indicated. It is still a valuable result considering the focus of the study being the coming century, but it needs to be highlighted that other studies derive to the same conclusion.**

**Response:** We changed the sentence to: "In agreement with Landschützer et al. (2018), also the model projections towards the end of this century demonstrate that the global amplification of $\delta$pCO$_2$ is due to the overall longterm increase of anthropogenic CO$_2$. A higher oceanic background CO$_2$ concentration enhances the effect of T-driven solubility changes on $\delta$pCO$_2$ and alters the seawater carbonate chemistry, also enhancing the DIC seasonality effect. "

**Comment 4: Page 9 lines 11-12: "Our results extend and refine the current views, in which the future amplification has been attributed uniquely to the DIC sensitivity". This is not correct. Both Landschützer et al. (2018)and Kwiatkowski and Orr (2018)**

discuss the attribution of other terms as well. The authors even briefly mention this in their introduction page 2 line 32: "Current literature suggests that the seasonal amplification is a consequence of an increase on the T and DIC contributions to pCO2 (Landschützer et al., 2018)..."

**Response:** We agree, we removed the sentence.

Comment 5: Page 9 lines 17-19: "The first complete analytical Taylor expansion of $pCO_2$ in terms of the variables DICs, TAs, T and S showed that DICs and T contributions are the main counteracting terms to control the $pCO_2$, both under present-day and future conditions. The prevalence of one term over the other in various regions remains similar, even under enhanced $CO_2$ conditions". This has also been shown by Landschützer et al. (2018) under past/present conditions, yet again this is not mentioned anywhere. Furthermore, by stating "The first complete Taylor expansion . . ." I suppose the authors mean within their own study, yet it created the impression that the authors refer to the first complete Taylor expansion overall, whereas, e.g. Kwiatkowski and Orr (2018) use the same Taylor expansion in their analysis.

**Response:** By "first complete analytical Taylor expansion, "we refer to the incorporation of T and S analytical terms, and therefore it is the first complete with analytical expressions in the four terms, which - to our knowledge- has not been done before. However, we agree this might be misleading, so we changed the sentence to: "The models confirm the well-established mechanisms controlling present-day $\delta pCO_2$ (Takahashi et al., 2002; Sarmiento and Gruber, 2006; Fay and McKinley, 2017). $DIC_s$ and T contributions are the main counteracting terms dominating the seasonal evolution of $\delta pCO_2$. Furthermore, the models show that under future conditions the controlling mechanisms remain unchanged. This result confirms the findings of Landschützer et al. (2018) that identified the same regional controlling mechanism for the past 30 years. The relative role of the DIC and T terms is regionally dependent. High latitudes and upwelling regions, such as the California Current system and the coast of Chile, are dominated by $DIC_s$ and the temperate low latitudes are driven by T. Only in the North Atlantic and North-Western Pacific the models show a dominance of thermal effects over non-thermal effects, which is in disagreement with observations. This further illustrates the urgent need for models to accurately represent regional oceanographic features to accurately reproduce the $\delta pCO_2$ characteristics. " The discussion on the difference between models and observations was added in the results and discussion section.

**Comment 6: Page 9 lines 23-26: "Spatially, we found that the magnitude of the contributions depends on the mean $pCO_2$ , its local sensitivities (DIC,TA,T,S) and the amplitude of their seasonal cycles ((DIC,TA,T,S)). The phases depend on the regional characteristics of the seasonal cycles and they moderate the counteracting nature of both contributions. The compensation of DICs and T contributions is most effective when they are six months out of phase." This mirrors again a conclusion drawn in Landschützer et al. (2018) (see e.g. Figure 3 in their study), whereas a comparison, discussion or even mentioning of this circumstance is missing here. Also regional characteristic have been discussed by Landschützer et al. (2018) and in terms of pH by Kwiatkowski and Orr (2018).**

**Response:** The sentence was changed to: "Moreover, the $pCO_2$ seasonal cycle amplitude depends on the relative magnitude and phase of the contributions. The models ensemble mean reproduces the highly effective compensation of $DIC_s$ and T contributions when they are six months out of phase, confirming previous studies (Takahashi et al., 2002; Landschützer et al., 2018). The compensation of DIC and T prevents a larger amplification of $\delta pCO_2$, even when both contributions are largely amplified."

**Comment 7: Another important result is only "hand wavy" introduced, namely that TA and S play a lesser role in the future pCO2 cycle amplification. One of the weak points of the Landschützer et al. (2018) study is that the authors ignore e.g. TA contributions, yet this study suggests that is of minor concern even when evaluating the century-long seasonal amplification. The authors also discuss second order terms here that have not been introduced in Landschützer et al. (2018) or Kwiatkowski and Orr (2018) , but this is also not mentioned/compared.**

**Response:** We added in the conclusions, page 11, line 6: "The amplification of the TA and S contributions have a small impact on $\delta pCO_2$ in most regions, except in the high latitudes where the TA contribution complements the DIC one, enhancing the non-thermal effect in this region. " We added in the results section, page 10, line 5: " In general, the $\Delta\delta T$ contribution gains importance as we move poleward in both hemispheres and therefore the second order terms originating from $\Delta\overline{pCO_2} \cdot \Delta\delta T$ also reinforce the amplification. Interestingly, in the high latitudes, the amplification through second order terms is as important as the change in the seasonality of the drivers. "

**Comment 8: Very interesting regional differences occur between the observation-based assessment of Landschützer et al. (2018) and this study, that are not discussed at all. Landschützer**

et al. (2018) find a DIC dominance in the high latitudes of both hemispheres, whereas the model based study suggests a T dominated increase in the high latitude northern hemisphere. Is this due to a model bias in seasonality. Is this the same across all models?.

**Response:** We added in the "Results and discussion" section, page 7, line 2: "The models show that the $\delta$pCO$_2$ in the 40ºN to 60ºN band is controlled by T, which disagrees with the above mentioned observations that show a non-temperature dominance in this band. The difference between models and observations arises from two regions: the North Atlantic basin and the North Western Pacific; specifically near the Oyashio Current, and the outflows from the Okhotsk Seas (see Supplementary Fig. S3). Most models show a T dominance in the North Atlantic basin; only CESM1-BGC and GFDL-ESM2M show a DIC dominance (see Supplementary Fig. S4). The North Atlantic is one of the major sinks of anthropogenic CO$_2$, however some models fail to estimate its uptake capacity (Goris et al., 2018). Goris et al. (2018) found that models with an efficient carbon sequestration present a DIC-dominated pCO$_2$ seasonal cycle in the North Atlantic, but models with low anthropogenic uptake show a T dominance in this region. In the North-Western Pacific, Mckinley et al. (2006) found that coarse models are not able to capture the intricate oceanographic features of this area, and therefore the pCO$_2$ seasonality is not well captured."

**Comment 9: The authors have conducted an extensive, interesting and certainly valuable analysis using state-of-the-art model outputs. Their methods are sound and their results nicely fit alongside the existing literature. The lack of discussion with the existing literature, however, is of major concern, particularly that the authors fail to acknowledge similar studies coming to the same conclusions. If the authors were to revise their manuscript and discuss their results in a fair way considering the existing literature, I believe this study can be considered for publication. The revisions however will affect the text throughout, hence I recommend major revisions of the manuscript.**

**Response:** As suggested by the referee, we have done a major revision of the manuscript. We thank again for the suggestions that helped to improve the manuscript; we placed our results and their relevance among the current literature and compared/contrasted our findings which previous results, in particular those by Landschützer et al. (2018).

**Comment 10: Abstract line 1: "observations" its observation-based**

**Response:** Changed to "observation-based results"

**Comment 11: Introduction page 1 line 22: a third of the anthropogenic $CO_2$ produced by fossil fuel burning, cement production and deforestation since the industrial revolution". The cited Sabine study suggest 48% since the beginning of industrialization. The referenced 1/3 refer to the annual uptake as stated in the second study cited, namely the Le Quere et al carbon budget.**

**Response:** We changed it to: "the ocean has absorbed nearly half of the anthropogenic $CO_2$ produced by fossil fuel burning and cement production since the industrial revolution (Sabine et al., 2004)"

**Comment 12: Page 2 line 21: [CO2(aq)]is introduced. For the non carbonate seawater chemists that read BG it would be helpful to explain the difference between [CO2] and [CO2(aq)]**

**Response:** We changed it to: " This is due to the ability of $CO_2$ to react with seawater to form bicarbonate [$HCO_3^-$] and carbonate [$CO_3^{2-}$], leaving only a small portion of the dissolved carbon dioxide in the form of aqueous $CO_2$ ([$CO_2(aq)$]). [$CO_2(aq)$] together with the carbonic acid ([$H_2CO_3$]) are defined as [$CO_2$]. Therefore, it is useful to define the total amount of carbon as DIC, which is the sum of the three carbon species ([$HCO_3^-$], [$CO_3^{2-}$] and [$CO_2$])."

**Comment 13: Page 4 line 11 and Supplement figure S1: The comparison between individual models gets worse in the high latitudes. Any idea why? The high latitude northern hemisphere is also where this study differs from the observation-based analysis of Landschützer et al. (2018).**

**Response:** In this figure we compare the $pCO_2$ amplification calculated as model output with the value from the Taylor expansion. The Taylor expansion is less precise in higher latitudes, probably because second order terms gain importance. The difference with Landschützer et al. (2018) was addressed in comment 8.

**Comment 14: Page 4 line 20, equation 3 and following: the delta terms also represent the mean seasonal cycle over 20 years (period 1 or period 2) hence they should have also an overbar (like the pCO2) for consistency.**

**Response:** We leave the nomenclature as it is, as by "mean" we refer to the mean value of the data, instead of the deviation of the mean, which is

the seasonal cycle.

**Comment 15: Page 5 line 14: "The range agrees with previous estimates by Takahashi et al. (2002)." Please add the comparison (visual or in table form), e.g. in the supplement for the readers of this study. Otherwise the reader has to jump around several different manuscripts for a simple comparison.**
**Response:** We added a supplementary figure S3, for better comparison with data from Takahashi et al. (2014), for a reference year 2005 and with Landschützer et al. (2017). We also added at page 5, after line 14 : "The ensemble mean initial seasonal amplitude range is in good agreement with observational estimates calculated for the reference year 2005 (Takahashi et al., 2014), and for the 1982-2015 period (Landschützer et al., 2017). The agreement between models and observations is remarkably good in the equatorial regions, but the initial amplitude is slightly overestimated in the mid and high latitudes (see Supplementary Fig. S3).The higher amplitude in models than observations is expected, as the initial period 2006-2026 already experienced an amplification compared to previous years. Moreover, Tjiputra et al. (2014) found that the ocean's $pCO_2$ historical trend is larger in models than observations when it is estimated in large scale areas of the ocean. However, they found that models' $pCO_2$ trends agree with observations when the trends are subsampled to the locations where the observations were taken, and therefore they do a good job reproducing well-known time series. Moreover, differences are expected as Pilcher et al. (2015) suggested that CMIP5 models perform well in reproducing the seasonal cycle timing, but still show considerable errors in reproducing the seasonal amplitude of $pCO_2$ due to differences in the mechanisms represented in each model, especially in subpolar biomes. "

**Comment 16: Page 5 line 21: "Our mean amplification factor estimation agrees with the lower end range of McNeil and Sasse (2016)." Please add numbers for the reader of this study.**
**Response:** We changed this sentence to: " Our mean amplification factor estimation agrees with the threefold amplification found for most of the ocean by McNeil and Sasse (2016)."

**Comment 17: Page 6 lines 8-9: "Our estimated contributions from DICs and T to the present day $pCO_2$ are in good agreement with the data based estimates (Takahashi et al., 2002; Fay and McKinley, 2017)." Please add a visual comparison or numbers for the readers of this study.**

**Response:** Instead of comparing with Takahashi et al. (2002), and Fay and McKinley (2017), we used the dataset of Takahashi et al. (2014) and calculated thermal and non thermal components for year 2005. We also added a comparison with the thermal and non-thermal components for years 1982-2015 that Peter Landschützer kindly provided to us. The results are shown in Supplementary Figure S3, and the discussion was added in section 3.2: "For most of the ocean, the ensemble mean estimated contributions from $DIC_s$ and T to the present-day $\delta pCO_2$ are in good agreement with the data-based estimates of Takahashi et al. (2014) and Landschützer et al. (2017), particularly in the equatorial regions (see Supplementary Fig. S3). However our T and DIC contributions are slightly larger in mid and high latitudes, for the same reasons the $pCO_2$ seasonal amplitude is overestimated (see Section 3.1). Also, differences arise between our $DIC_s$ contribution and the observation-based so called "non-thermal" contribution, because the non-thermal contribution also includes the total alkalinity and salinity effects. Nonetheless, between 40ºS-40ºN our ensemble mean shows that $\delta pCO_2$ is dominated by changes in temperature that control $CO_2$ solubility, which decreases in summer enhancing $pCO_2$, in agreement with observations. The Southern Ocean is controlled by DIC, that responds to changes in upwelling and phytoplankton blooms. Both mechanisms act together to decrease (increase) DIC in summer (winter) (Sarmiento and Gruber, 2006). " The discussion of northern high latitudes is added in comment 8.

**Comment 18: Page 7 lines 6-7: "DIC must not be confused with the Revelle factor, which is defined as R = DIC x gamma DIC". This statement comes a bit out of the blue and while true it is not clear to me why it appears here. Based on the equations/wording used in this study I don't see the danger that these terms are mixed up.**

**Response:** The Revelle factor and the sensitivity are different, and sometimes confused. We included the relationship because Takahashi et al. (1993) computed the Revelle factor. This sentence was rearranged as: "This follows the approach of Takahashi et al. (1993), however instead of computing the Revelle factor we use $\gamma_{DIC}$, both terms are related by R = DIC $\cdot$ $\gamma_{DIC}$."

**Other changes added:**

- page 7, line 25 we added: " In this region some models underestimate the $pCO_2$ trend (Tjiputra et al., 2014), and therefore the seasonal amplification might be underestimated too."

- page 7, line 32, we added: "Lower buffer factors (higher sensitivities factors) are found in regions where DIC and TA have similar values,

and they will decrease (increase) as the DIC/TA ratio in the oceans increases (Egleston et al., 2010). "

- page 8, line 3, we added: "$\gamma_{S_{fw}}$ decreases everywhere except in the Western Pacific Warm Pool. In this region $\gamma_{S_{fw}}$ increases probably due future changes in precipitation that enhance the fresh-water effect."

- page 9, line 5 was changed to: "Kwiatkowski and Orr (2018) demonstrated that the seasonality of the drivers is important to determine future changes in $[H^+]$ seasonality. In the same fashion, our results show that the four $\delta pCO_2$ drivers present changes in seasonality, and in particular $\delta DIC_s$ and $\delta T$ changes are important to explain future projections of the $\delta pCO_2$ amplitude."

**Response:** We did three changes based on this comment: We corrected the calculation and the sentence was changed to: " Using observations, Landschützer et al. (2018) found an increase of 2.2 $\mu$atm per decade, which is smaller than our findings of a total 42 $\mu$atm increase by the end of the century between 40°S-40°N, and a global-mean change of 81 $\mu$atm on the high latitudes. This difference is again possibly due the higher mean $pCO_2$ values in models than observations. " The discussion of the difference between models and observations was addressed in the response to Referee 1, Comment 15.

**Comment 4: Fig. 3/Fig. S1: These figures show (among other things) that the Taylor expansion generally does a good job in reproducing the actual $pCO_2$ calculated from model output. However, there seems to be an inconsistency between the two figures. Specifically, Fig.3 suggests that the Taylor expansion slightly overestimates the seasonal amplitude of $pCO_2$ (this is most evident for the 40S-70S latitude band), while Fig. S1 suggests exactly the opposite: an underestimation of the seasonal amplitude.**
**Response:** The labels of the S1 figure's axis were switched (x-axis label was y-axis, and vice versa). Figure S1 was corrected.

**Comment 5: p. 7, line 15: "decrease in the future to a global mean value of 0.035" .This number seems to be too small looking at Fig. 4c (middle column).**
**Response:** There was an error in the calculation. The sentence was changed to: "This value agrees with our global mean ensemble estimate of 0.0428. However, our analytical expression of $\gamma_T$ shows that this value varies regionally and, by reasons unknown to us, it might decrease in the future to a global mean value of 0.0415, (Fig. 4, row (c), third column). "

**Comment 6: p. 7, line 26: "with lower temperatures in winter and higher in summer" It might be good to clarify here that you do not mean lower temperatures in an absolute sense (i.e., winter temperatures are certainly projected to be higher at the end of century under RCP8.5 than they are at present).**
**Response:** This sentence was changed and we added the proper context: "All models show a slight increase in $\delta$T, only one model showed a slightly decrease in the southern region, and two models showed a decrease in the equatorial region during October to December. It is important to note that Fig. 5 shows the seasonal values, with the mean T removed. Therefore, when considering the positive T trends, the absolute summer values show an increase and the absolute winter values a decrease. This agrees with the results of Alexander et al. (2018); who showed that models project a seasonal intensification of T, with larger warm extremes and reduced cold extremes. The authors attributed the T seasonality intensification to an increased oceanic stratification and an overall shoaling of the mixed layer depth, which confines seasonal changes in a reduced volume of water, producing larger changes at the surface. They also showed that the intensification trends are stronger in summer than winter, as the mixed layer depth is shallower in summer. Moreover, ice covered regions will experience the largest increase in T seasonality due the loss of sea ice, because the ice melting/freezing moderates the surface water temperature seasonality(Carton et al., 2015). "

**Comment 7: p. 8, lines 26-27: "we decomposed the DICs and T contributions..." I only see the seasonal cycle and mean pCO$_2$ components in Fig. 6b, not the sensitivity component.**
**Response:** We changed the sentence to: "To further disentangle which of the two main drivers (DIC$_s$ or T) is most affected by $\Delta\overline{pCO_2}$, we decomposed the DIC$_s$ and T contributions in their sensitivity, seasonal cycle and $\overline{pCO_2}$ components. Figure 6, (b), shows the total DIC and T components

together with the $\Delta\overline{pCO_2}$ and seasonal cycles effects on them. The effects from the sensitivities are not depicted, as they only play a minor role. Only the $\Delta\gamma_{DIC}$ term gains importance in the Southern Ocean (not shown). "

**Comment 8: Technical corrections: 1) p. 1, line 2: Should be a rate of 2-3 muatm per decade?.**
Changed to decade.
**2) p. 3, line 5: "Methodology" misspelled.**
Corrected.
**3) p. 3, line 14: Should be scarce?.**
Changed to "scarced".
**4) p. 7, line 11: Should probably remove the word "change" here, since the annual cycle amplitude change is actually 168 muatm minus 96 muatm (i.e., 72 muatm).**
We removed the word "change" and changed the sentence to: "therefore, for a $\overline{pCO_2}$ equal to 800 $\mu$atm, the $\delta pCO_2$ amplitude due to $\delta DIC$ amounts to 168 $\mu$atm. "
**5) p. 7, line 15: Should be "row (c)".**
Changed to "row c)"

[revised manuscript text omitted]

15    Boden, T. A., Houghton, R. A., House, J. I., Keeling, R. F., Tans, P., Arneth, A., Bakker, D. C. E., Barbero, L., Bopp, L., Chang, J.,
Chevallier, F., Chini, L. P., Ciais, P., Fader, M., Feely, R. A., Gkritzalis, T., Harris, I., Hauck, J., Ilyina, T., Jain, A. K., Kato, E., Kitidis, V.,
Goldewijk, K. K., Koven, C., Landschützer, P., Lauvset, S. K., Lefevre, N., Lenton, A., Lima, I. D., Metzl, N., Millero, F., Munro, D. R.,
Murata, A., Nabel, J. E., Nakaoka, S., Nojiri, Y., O'Brien, K., Olsen, A., Ono, T., Perez, F. F., Pfeil, B., Pierrot, D., Poulter, B., Rehder,
G., Rödenbeck, C., Saito, S., Schuster, U., Schwinger, J., Séférian, R., Steinhoff, T., Stocker, B. D., Sutton, A. J., Takahashi, T., Tilbrook,
20    B., I. T. van der Laan-Luijkx, I. T., van der Werf, G. R., van Heuven, S., Vandemark, D., Viovy, N., Wiltshire, A., Zaehle, S., and Zeng,

[revised manuscript text omitted]

$$TA = \frac{K_1[CO_2]}{[H^+]} + 2\frac{K_1K_2[CO_2]}{[H^+]^2} + \frac{B_{tot}K_b}{(K_b + [H^+])} - [H^+] + \frac{K_w}{[H^+]} \tag{2}$$

Where $K_1$ and $K_2$ are defined as Millero et al. (2006), K$_w$ as Millero (1995) and K$_b$ according to Dickson (1990). From Eq.(1) we can obtain $[H^+]$ and from Eq.(2) we get $[CO_2]$ respectively as:

$$[H^+] = \frac{K_1[CO_2] + \sqrt{K_1^2[CO_2]^2 + 4K_1K_2[CO_2](DIC - [CO_2])}}{2(DIC - [CO_2])} \tag{3}$$

$$[CO_2] = \frac{[H^+]^2}{K_1[H^+] + 2K_1K_2}\left(TA - \frac{B_{tot}K_b}{(K_b + [H^+])} + [H^+] - \frac{K_w}{[H^+]}\right) \tag{4}$$

For $[H^+]$ the positive solution was chosen; the negative root gives a result far from real values. From Eq.(3) and Eq.(4) we can make a Talyor's expansion of $[H^+]$ and $[CO_2]$ respectively as:

$$\delta[H^+] = \left.\frac{\partial[H^+]}{\partial DIC}\right|_{\overline{CO_2},\overline{DIC},\overline{T},\overline{S}}\delta DIC + \left.\frac{\partial[H^+]}{\partial[CO_2]}\right|_{\overline{CO_2},\overline{DIC},\overline{T},\overline{S}}\delta[CO_2] + \left.\frac{\partial[H^+]}{\partial T}\right|_{\overline{CO_2},\overline{DIC},\overline{T},\overline{S}}\delta T + \left.\frac{\partial[H^+]}{\partial S}\right|_{\overline{CO_2},\overline{DIC},\overline{T},\overline{S}}\delta S \tag{5}$$

$$\delta[CO_2] = \left.\frac{\partial[CO_2]}{\partial TA}\right|_{\overline{TA},\overline{H},\overline{T},\overline{S}}\delta TA + \left.\frac{\partial[CO_2]}{\partial[H^+]}\right|_{\overline{TA},\overline{H},\overline{T},\overline{S}}\delta[H^+] + \left.\frac{\partial[CO_2]}{\partial T}\right|_{\overline{TA},\overline{H},\overline{T},\overline{S}}\delta T + \left.\frac{\partial[CO_2]}{\partial S}\right|_{\overline{TA},\overline{H},\overline{T},\overline{S}}\delta S \tag{6}$$

The overbars indicate the mean values of the variables in which the derivatives are evaluated. Finally, we insert $\delta[H^+]$ from Eq.(5) into Eq.(6), to get $[CO_2]$ in terms of DIC, TA, T and S:

$$
\begin{aligned}
\delta[CO_2] = &\left[1 - \left.\frac{\partial[CO_2]}{\partial[H^+]}\right|_{\overline{TA},\overline{H},\overline{T},\overline{S}}\left.\frac{\partial[H^+]}{\partial[CO_2]}\right|_{\overline{CO_2},\overline{DIC},\overline{T},\overline{S}}\right]^{-1} \cdot \left[\left.\frac{\partial[CO_2]}{\partial TA}\right|_{\overline{TA},\overline{H},\overline{T},\overline{S}}\delta TA\right.\\
&+ \left.\frac{\partial[CO_2]}{\partial[H^+]}\right|_{\overline{TA},\overline{H},\overline{T},\overline{S}}\left.\frac{\partial[H^+]}{\partial DIC}\right|_{\overline{CO_2},\overline{DIC},\overline{T},\overline{S}}\delta DIC\\
&+ \left(\left.\frac{\partial[CO_2]}{\partial T}\right|_{\overline{TA},\overline{H},\overline{T},\overline{S}} + \left.\frac{\partial[CO_2]}{\partial[H^+]}\right|_{\overline{TA},\overline{H},\overline{T},\overline{S}}\left.\frac{\partial[H^+]}{\partial T}\right|_{\overline{CO_2},\overline{DIC},\overline{T},\overline{S}}\right)\delta T\\
&\left.+ \left(\left.\frac{\partial[CO_2]}{\partial S}\right|_{\overline{TA},\overline{H},\overline{T},\overline{S}} + \left.\frac{\partial[CO_2]}{\partial[H^+]}\right|_{\overline{TA},\overline{H},\overline{T},\overline{S}}\left.\frac{\partial[H^+]}{\partial S}\right|_{\overline{CO_2},\overline{DIC},\overline{T},\overline{S}}\right)\delta S\right]
\end{aligned}
\tag{7}
$$

Comparing the terms from Eq.(7) to the desired Taylor's expansion:

$$\delta pCO_2 \approx \left.\frac{\partial pCO_2}{\partial DIC}\right|_{\overline{TA},\overline{DIC},\overline{T},\overline{S}}\delta DIC + \left.\frac{\partial pCO_2}{\partial TA}\right|_{\overline{TA},\overline{DIC},\overline{T},\overline{S}}\delta TA + \left.\frac{\partial pCO_2}{\partial T}\right|_{\overline{TA},\overline{DIC},\overline{T},\overline{S}}\delta T + \left.\frac{\partial pCO_2}{\partial S}\right|_{\overline{TA},\overline{DIC},\overline{T},\overline{S}}\delta S \tag{8}$$

We can identify the derivatives from Eq.(8), as follows:

$$\frac{\partial pCO_2}{\partial TA}\bigg|_{\substack{TA,DIC \\ \overline{T},\overline{S}}} = \overline{pCO_2} \cdot \frac{-\overline{TA_c}}{\overline{DIC} \cdot \Theta - \overline{TA_c^2}} \tag{9}$$

$$\frac{\partial pCO_2}{\partial DIC}\bigg|_{\substack{TA,DIC \\ \overline{T},\overline{S}}} = \overline{pCO_2} \cdot \frac{\Theta}{\overline{DIC} \cdot \Theta - \overline{TA_c^2}}$$

$$\frac{\partial pCO_2}{\partial T}\bigg|_{\substack{TA,DIC \\ \overline{T},\overline{S}}} = \overline{pCO_2} \cdot \frac{1}{\overline{DIC} \cdot \Theta - \overline{TA_c^2}}\left[\overline{TA_c} \cdot \left(\frac{\partial Alk_c}{\partial T} + \frac{\partial [B(OH)_4^-]}{\partial T} + \frac{\partial [OH^-]}{\partial T}\right) - \Theta \cdot \frac{\partial(DIC - [CO_2])}{\partial T}\right] - \frac{\overline{pCO_2} \cdot}{\overline{K_0}(T,S)}\frac{\partial K_0(T,S)}{\partial T}$$

5 $$\frac{\partial pCO_2}{\partial S}\bigg|_{\substack{TA,DIC \\ \overline{T},\overline{S}}} = \overline{pCO_2} \cdot \frac{1}{\overline{DIC} \cdot \Theta - \overline{TA_c^2}}\left[\overline{TA_c} \cdot \left(\frac{\partial \overline{TA_c}}{\partial S} + \frac{\partial [B(OH)_4^-]}{\partial S} + \frac{\partial [OH^-]}{\partial S}\right) - \Theta \cdot \frac{\partial(DIC - [CO_2])}{\partial S}\right] - \frac{\overline{pCO_2} \cdot}{\overline{K_0}(T,S)}\frac{\partial K_0(T,S)}{\partial S}$$

where $\Theta = [HCO_3^-] + 4[CO_3^{2-}] + \frac{[B(OH)_4^-][H^+]}{(k_b+[H^+])} + [H^+] + [OH^-]$ and $\overline{Alk}_c = [HCO_3^-] + 2[CO_3^{2-}]$. Below are some details of the specific concentrations derivatives.

$$\frac{\partial Alk_c}{\partial T,S} = \frac{[CO_2]}{[H^+]^2}\left(\frac{\partial k_1}{\partial T,S}[H^+] + 2k_1\frac{\partial k_2}{\partial T,S} + 2k_2\frac{\partial k_1}{\partial T,S}\right) \tag{10}$$

10 $$\frac{\partial(DIC - [CO_2])}{\partial T,S} = \frac{[CO_2]}{[H^+]^2}\left(\frac{\partial k_1}{\partial T,S}[H^+] + k_1\frac{\partial k_2}{\partial T,S} + k_2\frac{\partial k_1}{\partial T,S}\right)$$

$$\frac{\partial[B(OH)_4^-]}{\partial T} = \frac{B_{tot}[H^+]}{(k_b + [H^+])^2}\frac{\partial k_b}{\partial T}$$

$$\frac{\partial[B(OH)_4^-]}{\partial S} = \frac{B_{tot}[H^+]}{(k_b + [H^+])^2}\frac{\partial k_b}{\partial S} + \frac{k_b}{(kb + [H^+])}\frac{\partial B_{tot}}{\partial S}$$

$$\frac{\partial[OH^-]}{\partial T,S} = \frac{1}{[H^+]}\frac{\partial k_w}{\partial T,S}$$

[Figure]

**Figure S1.** pCO$_2$ seasonal cycle amplitude calculated from model output compared to its Taylor's expansion reconstruction in a) 2006-2026 and b) 2080-2100. Different colors indicate latitudinal ranges of zonal means, for the Atlantic (triangles), Pacific (circles) and Indian (stars) ocean basins. Large symbols represent the ensemble mean, and small symbols are the result for each model separately.

[Figure]

**Figure S2.** Column a) shows the simulated, ensemble-mean pCO$_2$ seasonal amplitude calculated as summer minus winter for each hemisphere respectively. b) to e) show DIC$_s$, T, TA$_s$ and S contributions to the pCO$_2$ summer-minus-winter amplitude. First and second rows represent respectively the 2006-2026 and 2080-2100 periods. Third row shows the difference between second and first rows. The amplitude was calculated from the climatology for periods 2006-2026 and 2080-2100.

[Figure]

**Figure S3.** Column a) shows the pCO$_2$ seasonal amplitude calculated as summer minus winter for each hemisphere respectively. b) and c) show the thermal and non-thermal contributions to pCO$_2$ seasonality respectively. First row shows CMIP5 models ensemble mean for the 2006-2026 period under the RCP8.5 scenario. Second row shows the estimates from Takahashi et al. (2014) dataset for a reference year 2005, with summer-minus-winter thermal and non-thermal contributions calculated as Takahashi et al. (2002). Third rows show the same components for the Landschützer et al. (2017) pCO$_2$ data-set, and the thermal and non-thermal estimations that Peter Landschützer facilitated us, for the period 1982-2015.

[Figure]

**Figure S4.** $\delta pCO_2$ climatology for column a) 2006-2026 and b) 2080-2100 periods calculated as in Fig. S2; each row is the result for a different model. Column c) shows the differences between column b) and a).

[Figure]

**Figure S5.** a) Ensemble zonal mean, zonal average of pCO₂ climatology and b) DIC contribution in color with overlying black contours of T contribution for 2006-2026 period. c) and d) same as a) and b) but for the 2080-2100 period.

[Figure]

**Figure S6. RCP8.5 ensemble zonal mean seasonal cycles:** a) $\delta TA_s$ and b) $\delta S$, for different latitudinal bands. Blue lines represent the 2006-2026 period, depicted for comparison with the 2080-2100 period shown by red lines. Different panels represent different latitudinal sections. $\delta TA_s$ is projected to slightly increase in all the bands, while $\delta S$ is projected to slightly decrease. The shading represents one standard deviation across the models.